# Bilateral JNK activation is a hallmark of interface surveillance and promotes elimination of aberrant cells

Deepti Prasad[1,2,3], Katharina Illek[1], Friedericke Fischer[1,3,4], Katrin Holstein[1], Anne-Kathrin Classen[1,3,5,6]*

[1]Hilde-Mangold-Haus, University of Freiburg, Freiburg, Germany; [2]Spemann Graduate School of Biology and Medicine (SGBM), University of Freiburg, Freiburg, Germany; [3]Faculty of Biology, University of Freiburg, Freiburg, Germany; [4]International Max Planck Research School for Immunobiology, Epigenetics, and Metabolism, Freiburg, Germany; [5]CIBSS Centre for Integrative Biological Signalling Studies, University of Freiburg, Freiburg, Germany; [6]BIOSS Centre for Biological Signalling Studies, University of Freiburg, Freiburg, Germany

*For correspondence:
anne.classen@biologie.uni-freiburg.de

**Competing interest:** The authors declare that no competing interests exist.

**Abstract** Tissue-intrinsic defense mechanisms eliminate aberrant cells from epithelia and thereby maintain the health of developing tissues or adult organisms. 'Interface surveillance' comprises one such distinct mechanism that specifically guards against aberrant cells which undergo inappropriate cell fate and differentiation programs. The cellular mechanisms which facilitate detection and elimination of these aberrant cells are currently unknown. We find that in *Drosophila* imaginal discs, clones of cells with inappropriate activation of cell fate programs induce bilateral JNK activation at clonal interfaces, where wild type and aberrant cells make contact. JNK activation is required to drive apoptotic elimination of interface cells. Importantly, JNK activity and apoptosis are highest in interface cells within small aberrant clones, which likely supports the successful elimination of aberrant cells when they arise. Our findings are consistent with a model where clone size affects the topology of interface contacts and thereby the strength of JNK activation in wild type and aberrant interface cells. Bilateral JNK activation is unique to 'interface surveillance' and is not observed in other tissue-intrinsic defense mechanisms, such as classical 'cell-cell competition'. Thus, bilateral JNK interface signaling provides an independent tissue-level mechanism to eliminate cells with inappropriate developmental fate but normal cellular fitness. Finally, oncogenic Ras-expressing clones activate 'interface surveillance' but evade elimination by bilateral JNK activation. Combined, our work establishes bilateral JNK interface signaling and interface apoptosis as a new hallmark of interface surveillance and highlights how oncogenic mutations evade tumor suppressor function encoded by this tissue-intrinsic surveillance system.

## Editor's evaluation

In this work, Prasad et al. show in *Drosophila* imaginal discs that a process of interface surveillance removes cells with inappropriate cell fate via activation of *JNK* and apoptosis. Importantly, *JNK* activity and apoptosis target most efficiently cells at the interface of small clusters of aberrant cells, but cells expressing oncogenic mutations are refractory to elimination, suggesting a mechanism for tumor formation. Therefore, this study is of interest to both the cell and developmental biology and the cancer biology fields.

## Introduction

Genetically altered cells appear in epithelial tissues at constant rate, either as a result of developmental errors or mutagenesis throughout adult life (*Starostik et al., 2020*; *Martincorena, 2019*). Surveillance and removal of genetically altered cell are required to maintain tissue and organismal health. In addition to immune-cell-dependent processes (*Galli et al., 2020*; *Hiam-Galvez et al., 2021*; *Mahapatro et al., 2021*), tissue-intrinsic mechanisms, such as *cell-cell competition*, *epithelial defense against cancer* (EDAC), or *interface contractility*, have been identified to remove aberrant cells from tissues (*Matamoro-Vidal and Levayer, 2019*; *Tanimura and Fujita, 2020*; *Macara et al., 2014*; *Merino et al., 2016*; *Baker, 2020*; *Bielmeier et al., 2016*). In classical cell-cell competition scenarios, the comparison of cell fitness between neighboring cells is the driving force of cell elimination, with less fit 'loser' cells being eliminated by fit 'winner' cells (*Merino et al., 2016*; *Baker, 2020*; *Levayer and Moreno, 2013*; *Baumgartner et al., 2021*; *Kucinski et al., 2017*). Mutations that interfere with house-keeping functions, such as proteostasis, cellular metabolism, or genome maintenance, emerged as drivers of cell-cell competition (*Baumgartner et al., 2021*; *Blanco et al., 2020*; *Ji et al., 2019*; *Baker et al., 2019*; *Lee et al., 2018*; *Baillon et al., 2018*; *Kale et al., 2018*; *Ochi et al., 2021*).

In contrast, mutations may also interfere with signaling pathways or transcriptional networks that set up specific cell fate and differentiation programs (*Hiremath et al., 2022*; *Stuelten and Zhang, 2021*; *Brumbaugh et al., 2019*). These mutations may change a cell's fate and developmental trajectory but, importantly, may not disrupt cellular fitness per se. In the absence of fitness information, comparison of fitness states cannot be used to eliminate this aberrant cell. Thus, inappropriately specified cells create a distinct challenge to tissue health and, consequently, must activate a distinct program for their detection and elimination from tissue.

We previously established *interface contractility* as a novel paradigm for tissue-intrinsic defense mechanisms against aberrant, misspecified cells in *Drosophila* imaginal discs (*Bielmeier et al., 2016*). To reflect the more general implications of this mechanism, we will from now on use the term *'interface surveillance'*. Briefly, we demonstrated that two adjacent cells appear to be able to compare their fate and differentiation status: cells respond to pronounced fate differences between them by recruiting actomyosin to their shared contact surfaces. This response drives cell segregation between two clonal cell populations via smoothening of the contractile clone interface and is accompanied by elevated apoptosis (*Bielmeier et al., 2016*). This is a surprisingly universal response to relative differences in cell fate and differentiation states. Mosaic manipulation of the patterning pathways Dpp/TGF-β, Wg/WNT, Hh/Shh, JAK/STAT, and Notch or cell-fate-specifying transcription factors (Abd-B, Arm, APC, Iro-C, Omb, Yki, En/Inv, Ap, and Ci) induces actomyosin recruitment, clone smoothening, and apoptosis in imaginal discs (*Gibson and Perrimon, 2005*; *Shen and Dahmann, 2005*; *Widmann and Dahmann, 2009b*; *Widmann and Dahmann, 2009a*; *Pallavi et al., 2012*; *Gandille et al., 2010*; *Bessa et al., 2009*; *Aldaz et al., 2005*; *Beuchle et al., 2001*; *Prober and Edgar, 2000*; *Liu et al., 2000*; *Worley et al., 2013*; *Perea et al., 2013*; *Gold and Brand, 2014*; *Classen et al., 2009*; *Bell and Thompson, 2014*; *Organista and De Celis, 2013*; *Villa-Cuesta et al., 2007*; *Shen et al., 2010*). Importantly, these responses are induced in a strict position-dependent manner according to the cell fate of the surrounding cells. For example, Cubitus interruptus (Ci)-expressing clones have normal corrugated shapes in anterior wing compartments, where Ci-activation by Hedgehog (Hh) signaling is high. However, Ci-expressing clones undergo clone smoothening and die in posterior compartments, where Ci-signaling is normally low (*Bielmeier et al., 2016*). In summary, the characteristic contractile interface response observed in interface surveillance is exclusively driven by differences in cell fate programs between neighboring cells.

Interface surveillance efficiently eliminates aberrantly misspecified cells in the absence of fitness differences and thus in the absence of information about which of two neighboring cells should survive. Instead, by comparison to the program of the surrounding cells, interface surveillance uses spatial context and defines a clone as 'different'. As a consequence, not only aberrant clones are eliminated from the tissue but remaining wild-type clones are also eliminated when surrounded by a majority of aberrant cells (*Bielmeier et al., 2016*). Thus interface surveillance promotes apoptosis of cells, which appear aberrant by their relative mispositioning with respect to the fate of surrounding cells. This distinguishes interface surveillance from cell-cell competition, which ensures that always the 'aberrant' loser cell dies, independent of spatial context (*Figure 1—figure supplement 1*). Ultimately, elimination of cells by interface surveillance depends on clone size. Single aberrant cells and small

aberrant clones are more efficiently eliminated. Moreover, small clones present with higher level of apoptosis than larger clones (*Bielmeier et al., 2016*). We previously suggested that this may depend on compression induced by the contractile interface around clones, which theoretically gives rise to stronger compression of clones with a small circumference. However, this hypothesis or alternative pathways have not been experimentally tested. In fact, the molecular and cellular pathways, which drive elimination by interface surveillance are not known.

Using *Drosophila* wing imaginal disc model, we describe here a surprisingly ingenious solution to remove aberrant cells by interface surveillance. We find that interface surveillance is associated with activation of pro-apoptotic JNK signaling in cells at the interface of two differently fated cell populations, invoking a rejection response between two very different yet seemingly egalitarian cell types. This bilateral response of two contacting cell types drives apoptosis at clonal interfaces. JNK activation scales with clone size and thereby correlates with efficient elimination of single aberrant cells and small clones. This is consistent with a model where cell contact topologies at the interface of large and small clones differ, resulting in size-dependent apoptotic susceptibility.

## Results
### Apoptosis is essential to eliminate cells by interface surveillance

To better understand how interface surveillance drives elimination of aberrant cells, we analyzed mosaic wing imaginal discs using two genotype classes affecting cell fate: (1) cell fate regulators that are not normally expressed in the wing disc and (2) those that are expressed but in specific spatial patterns. The first class is represented by the transcription factors Fkh and Ey, which are master regulators of salivary gland (*Andrew et al., 2000*; *Haberman et al., 2003*) and eye specification (*Halder et al., 1995*), respectively. These genes are not expressed in the wing discs, and their ectopic expression causes interface surveillance hallmarks like actomyosin enrichment at clone interface in any position of the wing disc (*Figure 1A–E''*; *Figure 1—figure supplement 2A-D', J-K''*). The second class is represented by Dpp-mediated (Tkv), EGF-mediated (Egfr), Wg-mediated (Arm), and Hh-mediated (Ci) patterning programs. Dpp-, EGF-, and Wg-signaling are predominantly active in the central wing pouch, whereas Hh-signaling is activated in the anterior wing compartment (*Figure 1—figure supplement 2E-I*). Consequently, ectopic activation of these pathways by expression of $tkv^{CA}$, $Egfr^{CA}$, or $arm^{S10}$ causes actomyosin enrichment and clone smoothening in lateral pouch domains (*Figure 1F–G''*, *Figure 1—figure supplement 2L-O''*). In contrast, actomyosin enrichment and clone smoothening are detected in the posterior compartment upon expression of *ci* (*Figure 1—figure supplement 2P-Q''*). Consistent with the idea that it is specifically the difference between neighboring cell fates which induce interface surveillance, we found that loss of *tkv*-function induces actomyosin enrichment and clone smoothening in domains where Dpp-signaling is usually high (*Figure 1H–I''*). Previous studies similarly demonstrate pattern-specific clone smoothening for mosaic clones with loss-of-function mutations in *vg* (*Widmann and Dahmann, 2009b*), Iro-C (*Villa-Cuesta et al., 2007*), *omb* (*Shen et al., 2010*), *ci* (*Dahmann and Basler, 2000*), *ap* (*Klipa and Hamaratoglu, 2019*), *en* (*Zecca et al., 1995*), or Polycomb complexes (*Beuchle et al., 2001*). Combined, these data highlight that aberrant cells, as identified by relative mispositioning with respect to the fate of surrounding cells, induce interface surveillance hallmarks.

As the elimination of clones by interface surveillance is potentially driven by apoptosis (*Bielmeier et al., 2016*), we analyzed the distribution of apoptotic patterns in all genotypes in more detail. We observed that apoptosis of *fkh*-, $tkv^{CA}$-, and *tkv-RNAi*-expressing clones specifically occurred in positions, where also interface surveillance hallmarks, such as clone smoothening and actomyosin enrichment, occurred (*Figure 1J–M and Q*). Similarly, *ci*-expressing clones are specifically eliminated from posterior compartments. Moreover, as reported previously, small aberrant clones are activating apoptosis more efficiently than larger ones (*Figure 1R*, *Figure 1—figure supplement 3A*; *Bielmeier et al., 2016*). Combined, we conclude that elimination of aberrant cells strongly correlates with interface responses and likely proceeds via apoptosis.

To confirm that apoptosis was generally required to eliminate aberrant cells from the tissue, we expressed *Diap1* or *p35*, which are inhibitors of initiator and executioner caspases (i.e. Dronc or Dcp1), in *fkh*- and *ey*-expressing clones. Indeed, these *fkh*- and *ey*-expressing clones survived and remained integrated in the epithelial layer (*Figure 1N–P and S*; *Figure 1—figure supplement 3B-F*).

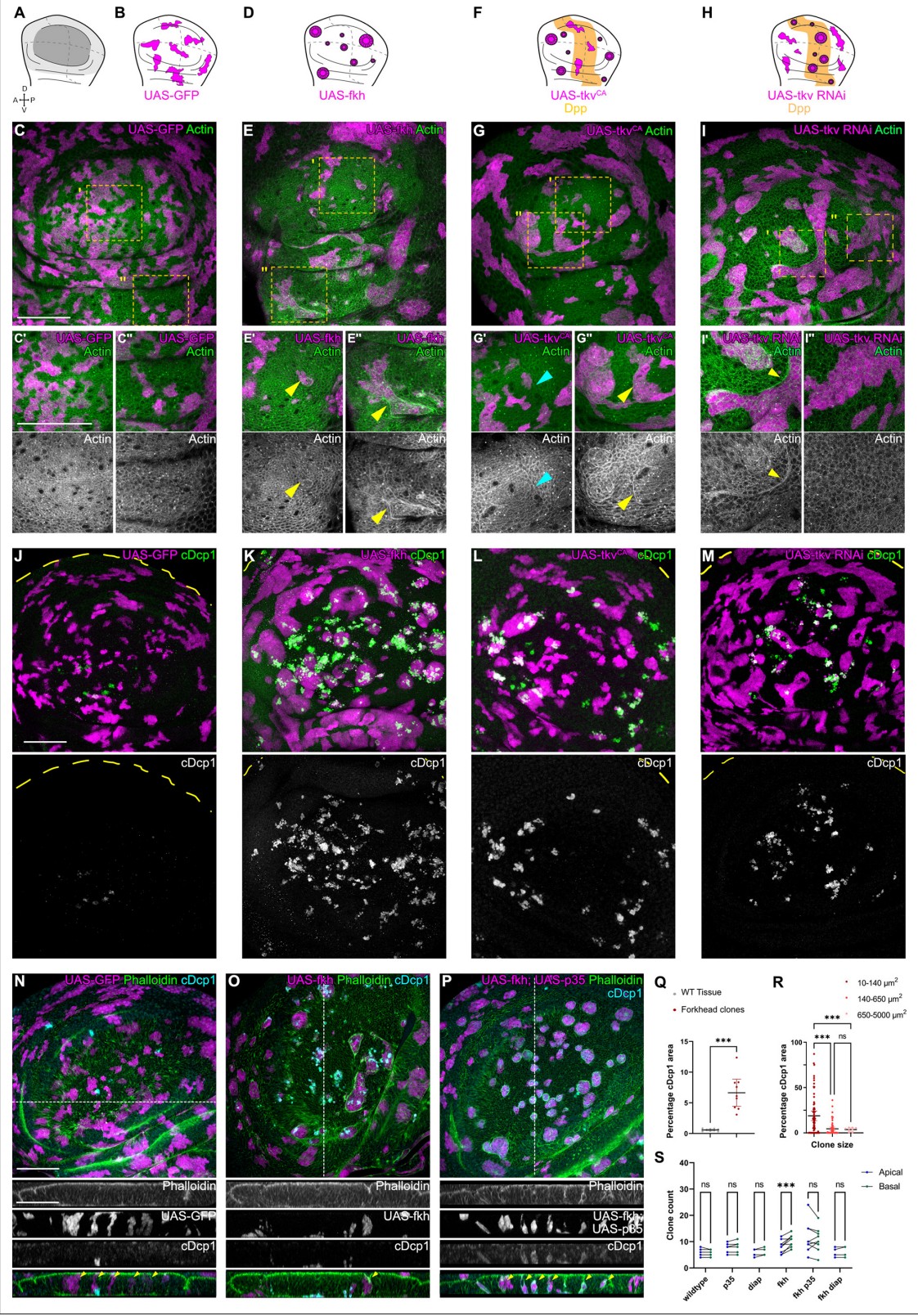

**Figure 1.** Apoptosis is essential to eliminate cells by interface surveillance. (**A**) Wing disc illustration highlighting the pouch (dark gray), hinge (light gray), and notum (white), as well as characteristic folds (continuous lines) and dorso-ventral or anterior-posterior compartment boundaries (dashed lines) (**A**). (**B, D, F, and H**) Wing discs schemes illustrating patterns of Dpp-signaling (**F and H**) (orange represents endogenous expression or activity, white represents lack thereof). Clones (magenta) that only express GFP (**B**) or whose fate is like that of surrounding cells (magenta clones in orange domains in

*Figure 1 continued on next page*

*Figure 1 continued*

F, or in white domain in H) do not induce interface surveillance and maintain irregular clone shapes. Clones whose fate is different to that of surrounding cells because of altered cell fate and differentiation programs (magenta clones in white domains in D, F and in orange domains in H) activate interface surveillance and thus experience interface smoothening. In extreme cases clones form cysts, as the apical surface buckles from interface contractility-induced compression. Clones express *GFP* (**B**), *fkh* (**D**), *tkv*$^{CA}$, (**F**) or *tkv*-RNAi (**H**). (**C, E, G, and I**) Wing disc carrying mosaic clones (magenta) expressing *GFP* (**C**), *fkh* (**E**), *tkv*$^{CA}$ (**G**), or *tkv*-RNAi (**I**) stained with phalloidin to visualize Actin (green or gray). Yellow frames mark regions in pouch center (**C′, E′, G′, and I′**), pouch periphery (**G″ and I″**), and hinge (**C″ and E″**). Note that these frames distinguish domains of high and low endogenous Dpp-activity in **F and H**, respectively. Cyan arrows point to accumulation of actin normally observed in all third instar wing discs at the A-P compartment boundary (**G′**). Yellow arrows point to apical enrichment of actin at clone boundaries. (**E′**) focusses on the apical region of a clone that has undergone buckling. (**J–M**) Maximum-intensity projections of basal sections of wing discs carrying mosaic clones (magenta) expressing *GFP* (**J**), *fkh* (**K**), *tkv*$^{CA}$ (**L**), and *tkv*-RNAi (**M**), stained for cleaved Dcp1 (cDcp1) to visualize apoptosis (green or gray). Please refer to (**F and H**) for endogenous activation patterns of Dpp signaling in wing discs. (**N–P**) Maximum-intensity projections of basal sections of wing discs carrying mosaic clones (magenta or gray) expressing *GFP* (**N**), *fkh* (**O**), or *fkh,p35* (**P**), stained with phalloidin to visualize Actin (green or gray) or stained for cleaved Dcp1 (cDcp1) to visualize apoptosis (cyan or gray). Dashed white lines indicate position of cross-sections shown below. Yellow arrowheads in the cross-section overlays indicate viable clones still integrated in the epithelium. (**Q**) Quantification of the relative percentage of apoptotic area in *fkh*-expressing clones, as compared to surrounding wild type tissue. Graph displays mean ± 95% CI. n=9 wing discs each. See *Figure 1—source data 1*. (**R**) Quantification of the relative percentage of apoptotic area after binning into different clone sizes. Dark red (small clones, 10–140 µm$^2$, approximately containing 1–6 cells), red (medium clones, 140–650 µm$^2$, approximately containing 7–35 cells), and pink (large clones, 650–5000 µm$^2$, approximately containing above 35 cells) in discs with *fkh*-expressing clones. n=140 clones from n=6 wing discs. See *Figure 1—source data 2*. (**S**) Quantification of the number of clones detected apically (blue) or basally (green) in the wing disc pouch, where clones express either *GFP; p35; Diap1; fkh; fkh, p35,* or *fkh, Diap1*. The numbers of wing discs analyzed per genotype include GFP (9); *p35* (9); *Diap1*(5); *fkh* (9); *fkh, p35* (9); and *fkh, Diap1* (4). See *Figure 1—source data 3*. (**Q, R, and S**) Paired Student's t-tests (**Q**) and one-way ANOVA tests (**R and S**) were performed to test for statistical significance, ns = not significant, ***p≤0.001. Scale bars = 50 µm.

The online version of this article includes the following source data and figure supplement(s) for figure 1:

**Source data 1.** for *Figure 1Q*.

**Source data 2.** for *Figure 1R*.

**Source data 3.** for *Figure 1S*.

**Figure supplement 1.** Apoptosis is essential to eliminate cells by interface surveillance.

**Figure supplement 2.** Apoptosis is essential to eliminate cells by interface surveillance.

**Figure supplement 3.** Apoptosis is essential to eliminate cells by interface surveillance.

**Figure supplement 3—source data 1.** for *Figure 1—figure supplement 3A*.

**Figure supplement 3—source data 2.** for *Figure 1—figure supplement 3E*.

**Figure supplement 3—source data 3.** for *Figure 1—figure supplement 3F*.

Importantly, we did not observe viable delaminated cells, indicating that mechanically driven live cell extrusion, as described for mammalian EDAC mechanisms, is not relevant for cell elimination in imaginal discs (*Tanimura and Fujita, 2020*; *Kon and Fujita, 2021*). Importantly, inhibition of apoptosis did not interfere with interface responses, such as clone smoothening, as reported previously (*Bielmeier et al., 2016*; *Shen and Dahmann, 2005*; *Widmann and Dahmann, 2009b*). These results reveal that recruitment of actomyosin to the clonal interface and thus activation of interface surveillance per se occur upstream and independent of apoptosis. Combined, these results demonstrate that interface surveillance activates apoptotic pathways to eliminate aberrant cells.

## Interface surveillance activates bilateral JNK interface signaling

To understand which signals drive apoptosis in aberrant cells, we analyzed signaling through JNK/AP-1, a potent tissue stress response pathway with important functions in cell death decision (*Pinal et al., 2019*; *Dhanasekaran and Reddy, 2017*; *Niethammer, 2016*). We found JNK to be consistently activated but, surprisingly, in a striking bilateral pattern at clonal interfaces. Specifically, in *fkh-* and *ey*-expressing clones, a band corresponding to about one cell-row on each side of the interface displayed the strongest activation of the JNK-reporter *TRE-RFP* (*Figure 2A–C1 and J*; *Figure 2—figure supplement 1A, B*; *Chatterjee and Bohmann, 2012*). The surrounding wild-type cells allowed us to analyze a spatial domain representing a second cell-row adjacent to wild-type interface cells. Yet, this second band, which was one step removed from the interface contacts, already showed a significantly lower activation of JNK. The JNK-activity reporter *puc-LacZ*, while less sensitive, independently confirmed that the observed patterns are JNK-specific and occur bilaterally at clonal interfaces (*Figure 2—figure*

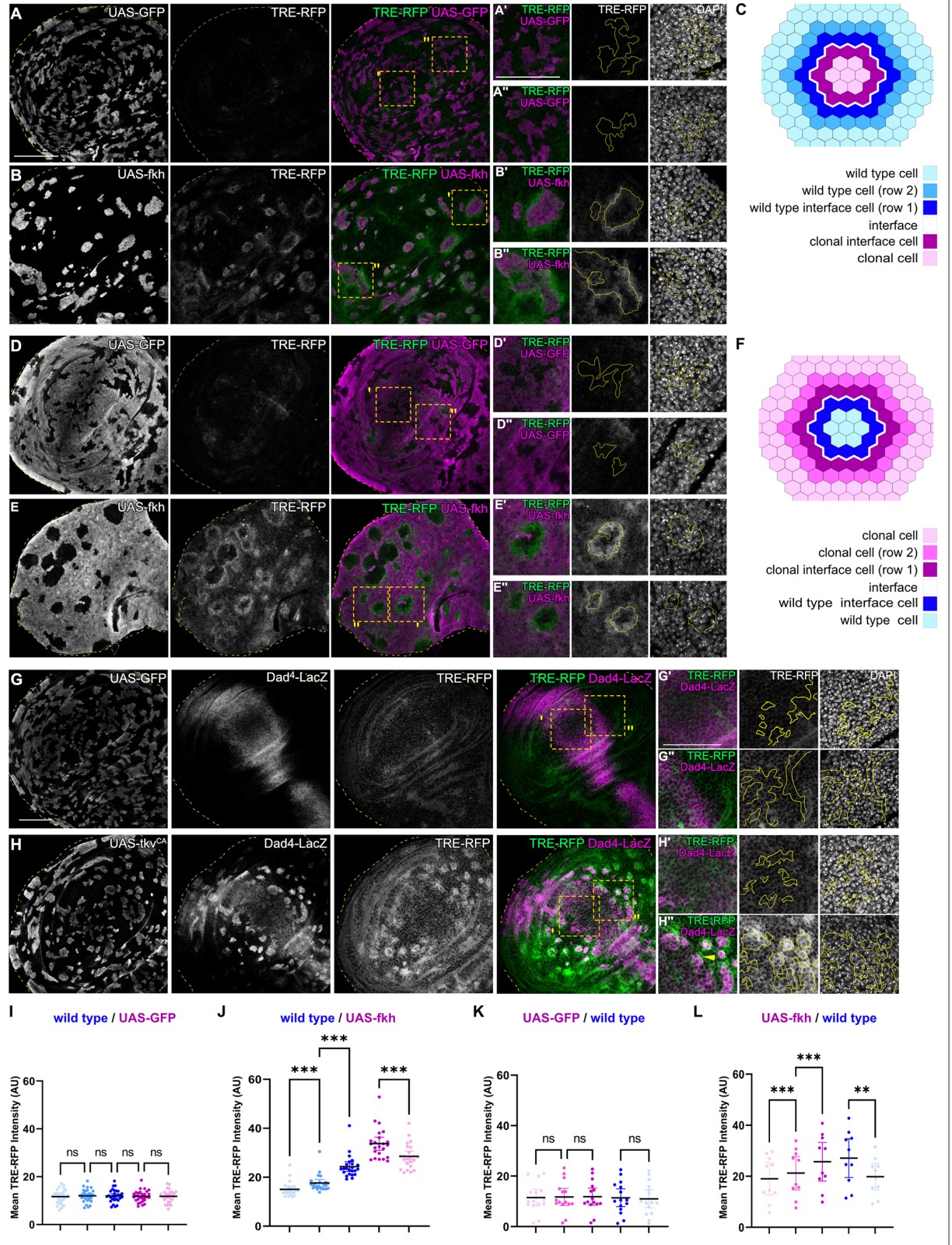

**Figure 2.** Interface surveillance activates bilateral JNK interface signaling. (**A, B, D, and E**) Wing discs carrying mosaic clones (magenta or gray) expressing *GFP* (**A and B**) or *fkh* (**D and E**), as well as the JNK reporter TRE-RFP (green or gray). Discs were stained with DAPI to visualize nuclei (gray). Dotted yellow lines mark wing disc boundaries. Yellow frames mark regions shown in (**A', A'', B', B'', D', D'', E', and E''**). Continuous yellow line in (' and '') panels marks clone boundaries (i.e. 'interface'). Different length of heat shock induction allows scaling of clone sizes from minority into majority

*Figure 2 continued on next page*

*Figure 2 continued*

topologies. (**C and F**) Schemes depicting specific zones in and around clones that were quantified. Cell rows (row 1, row 2, and interface cells) were approximated as 4 µm bands (see Materials and methods and **Figure 5—figure supplement 1A** for details). The interface is the cell contact between both genotypes. Due to spatial limitations, 'row 2' cells could only be analyzed in the majority/surrounding genotype. (**G and H**) Wing disc carrying mosaic clones (gray) expressing *GFP* (**G**) or *tkv*$^{CA}$ (**H**) and the Dpp reporter *Dad*$^4$-*LacZ* (gray or magenta) and the JNK reporter TRE-RFP (gray or green). Discs were stained with DAPI to visualize nuclei (gray). Dotted yellow lines mark wing disc boundaries. Yellow frames mark regions shown in (**G', G''**, **H', and H''**). *Dad*-*LacZ* is induced by *tkv*$^{CA}$-expression, revealing where *tkv*$^{CA}$-expressing clones are like their surroundings and where they are not. Continuous yellow line in (' and '') panels marks clone boundaries. Yellow arrow highlights TRE-RFP activation by *tkv*$^{CA}$-expressing clones right at the boundary of endogenous Dpp-signaling. (**I, J, K, and L**) Quantifications of TRE-RFP intensities for experiments shown in (**A–E**) within specific zones of clones expressing *GFP* (**I and K**) or *fkh* (**J and L**). Note that in (**K and L**) GFP-negative wild-type clones are amidst GFP-expressing wild-type cells (**K**) or amidst *fkh*-expressing cells (**L**). Please refer to (**B**) as legend for (**I and J**) and refer to (**E**) as legend for (**K and L**). Graphs display mean ± 95% CI. One-way ANOVA tests were performed to test for statistical significance, ns = not significant, ***p≤0.001. Wing discs analyzed in each genotype include n=30 (**I**), n=23 (**J**), n=15 (**K**), and n=10 (**L**). See *Figure 2—source data 1*; *Figure 2—source data 2*; *Figure 2—source data 3*; *Figure 2—source data 4*. Scale bars = 50 µm.

The online version of this article includes the following source data and figure supplement(s) for figure 2:

**Source data 1.** for *Figure 2I*.

**Source data 2.** for *Figure 2J*.

**Source data 3.** for *Figure 2K*.

**Source data 4.** for *Figure 2L*.

**Figure supplement 1.** JNK interface signaling is a robust hallmark of interface surveillance responses.

**Figure supplement 1—source data 1.** for *Figure 2—figure supplement 1B*.

**Figure supplement 1—source data 2.** for *Figure 2—figure supplement 1D*.

**Figure supplement 2.** Interface JNK signaling can be detected using the *puc*-LacZ reporter.

*supplement 2*). Combined, this demonstrates that both wild type and aberrant cells activate JNK signaling specifically when in contact with each other.

Consistent with the idea that interface surveillance strictly responds to fate differences between neighboring cell, we also observed bilateral JNK activity when *fkh*- or *ey*-expressing cells represented the majority population in the tissue and enclosed small islands of remaining wild type cells. Importantly, the large area of *fkh*- or *ey*-expressing cells in this experiment now allowed us to spatially analyze signaling in a band representing another cell-row adjacent to *fkh*- or *ey*-expressing interface cells. Similar to the second row band in wild-type cells discussed above, cells adjacent to *fkh*- or *ey*-expressing interface cells also displayed a significantly lower activation of JNK (*Figure 2D–F, K and L*; *Figure 2—figure supplement 1C-D*).

Importantly, bilateral JNK interface responses could also be observed in discs carrying *tkv*$^{CA}$, *Egfr*$^{CA}$, *arm*$^{S12}$, and *ci*-expressing clones. Here, however, JNK interface signaling strictly correlated with position of clones within the respective Dpp-, EGF-, Wg-, and Hh-patterning fields, confirming that JNK interface signaling responds to cell fate differences between neighboring cells (*Figure 2G–H''*; *Figure 2—figure supplement 1E-J*). While the resolution and strength of JNK activation may be affected by cellular dynamics, such as survival of clones in different wing disc domains, all areas of the wing disc can respond with JNK interface activation to the presence of differently fated cells. Combined, these observations establish JNK interface signaling as novel hallmark of interface surveillance and highlight the activation of a central stress signaling pathway upon direct cell surface contact between distinct cell fates.

## JNK activation is not dependent on JNK activation in the adjacent interface cell

We wanted to understand if bilateral JNK activation is a cell-autonomous response induced by contact between different cell fates. To begin to address this question, we first reduced JNK signaling within one cell type by co-expressing a dominant-negative *bsk* (*bsk*$^{DN}$) construct in cells that also express *ey* or *tkv*$^{CA}$. *bsk*$^{DN}$ is a potent suppressor of JNK signaling and completely abolished activity of JNK in wild-type clones and in *ey*- or *tkv*$^{CA}$-expressing clones, as judged by complete lack of intra-clonal TRE-RFP reporter activity (*Figure 3A and C*, *Figure 3—figure supplement 1D*; *Cosolo et al., 2019*). We found that eliminating JNK signaling in the *ey*-expressing interface did not prevent JNK activation

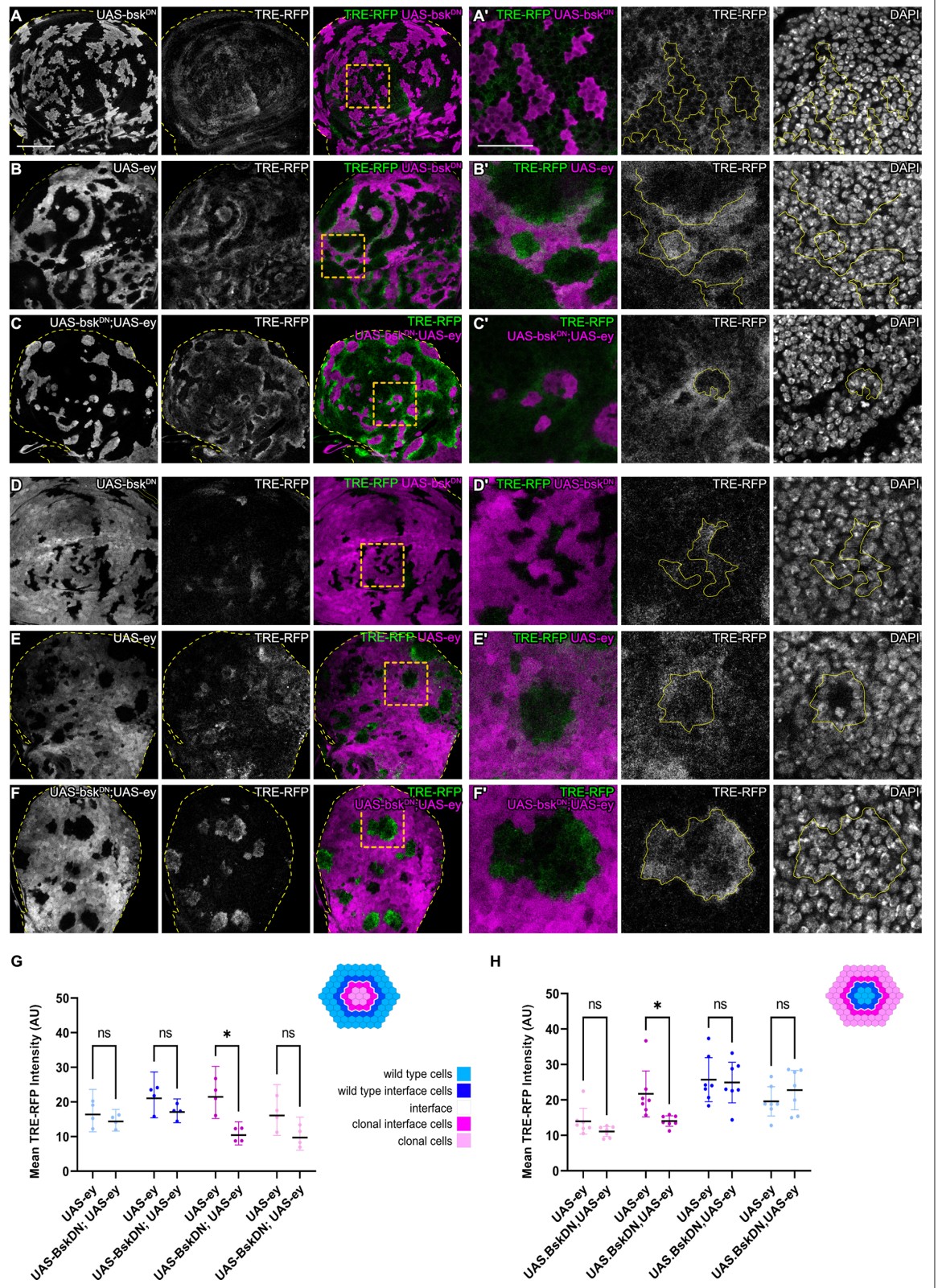

**Figure 3.** JNK activation is not dependent on JNK activation in the adjacent interface cell. (**A–F**) Wing discs carrying mosaic clones (magenta or gray) expressing *Bsk*$^{DN}$ (**A and D**), *Ey* (**B and E**), or *Bsk*$^{DN}$, *Ey* (**C and F**) and expressing the JNK reporter TRE-RFP (gray or green). Discs were stained with DAPI to visualize individual nuclei (gray). Yellow dashed lines in (**A–F**) demarcate the wing disc outline. Yellow frames mark regions shown in (') panels. Continuous yellow lines in (') panels demarcates clone boundaries (i.e. interface). (**G and H**) Quantifications comparing the TRE-RFP intensities in specific

*Figure 3 continued on next page*

*Figure 3 continued*

zones of clones expressing *Ey* or *Bsk^DN*, *Ey* (**G and H**). Graph in (**G**) represent experiments shown in (**A–C**) where clones are surrounded by wild-type cells and graph in (**H**) represent experiments shown in (**D–F**) wild-type cells are surrounded by cells with *Ey* or *Bsk^DN*, *Ey* genotypes. (**H**) Graphs display mean ± 95% CI. Two-way ANOVA tests were performed to test for statistical significance, ns = not significant, *p≤0.05. n=4 wing discs per genotype in (**G**), and n=7 wing discs per genotype in (**H**). See *Figure 3—source data 1*; *Figure 3—source data 2*. Scale bar = 50 µm.

The online version of this article includes the following source data and figure supplement(s) for figure 3:

**Source data 1.** for *Figure 3G*.

**Source data 2.** for *Figure 3H*.

**Figure supplement 1.** JNK activation is not dependent on JNK activation in the adjacent interface cell.

in neighboring wild-type cells, irrespective of whether these wild-type cells surrounded *ey*-expressing cells or were being surrounded by them (*Figure 3*). Similarly, expression of *bsk^DN* did not prevent the pattern-specific activation of JNK in wild-type cell at the interface with *tkv^CA*-expressing clones (*Figure 3—figure supplement 1*). These experiments demonstrate that contact between different cell fates activates JNK signaling on each side of the interface – independent of JNK activity in the neighboring interface cell. While these experiments only test if JNK activation in one interface cell is dependent on JNK activation in the neighboring interface cell, our results point to a model where JNK activation is a cell-autonomous response induced by contact between differently fated cells. Cell-autonomous JNK activation invokes independent stress responses to a cell fate difference within two differently fated yet equally viable cells.

## JNK interface signaling is not activated in classical cell-cell competition models, and is not required for interface actomyosin enrichment

To understand if JNK signaling is a specific hallmark of interface surveillance, we analyzed JNK-reporter activity in mosaic tissues subject to classical cell-cell competition. We specifically analyzed three well-established cell-cell competition genotypes: (1) clones heterozygous mutant for *RpS3*, representing a loser genotype (*Kale et al., 2015*), and two genotypes representing super-competitive winners in form of clones, (2) ectopically expressing *Dmyc* (*de la Cova et al., 2004*; *Moreno and Basler, 2004*), and (3) with reduced levels of the Hippo/Yorkie pathway component *wts* (*Tyler et al., 2007*). Importantly, none of these three genotypes displayed JNK interface signaling (*Figure 4*, *Figure 4—figure supplement 1*). Thus, while JNK activity may be observed cell-autonomously in loser genotypes due to disruption of cellular homeostasis (*Kucinski et al., 2017*), classical cell-cell competition models related to proteostasis and survival signaling do not induce contact-dependent JNK signaling at the clonal interface. These results strongly demonstrate that JNK interface signaling is a specific hallmark of interface surveillance and confirm that interface surveillance is a tissue-intrinsic error correction program distinct from classical cell-cell competition.

Another interface surveillance hallmark that is not observed in classical cell-cell competition is the enrichment of actomyosin at clonal interfaces and the resulting clone smoothening. The co-occurrence of JNK signaling and actomyosin enrichment at the clonal interface and the fact that JNK can be an upstream regulator of the actin dynamics (*Külshammer and Uhlirova, 2013*; *Brock et al., 2012*; *Kwon et al., 2010*) led us to ask if JNK interface signaling causes actomyosin enrichment at clonal interfaces. To test this, we first reduced JNK activity within clones by co-expression of a dominant-negative *bsk* (*bsk^DN*) construct. However, the interfaces of *ey*, *bsk^DN* or *tkv^CA*, *bsk^DN* co-expressing clones remained smooth, and actin enrichment could still be observed (*Figure 4—figure supplement 2A-D'*). To more directly test the contribution of JNK interface signaling on both sides of the clonal interface, we expressed *bsk^DN* in the entire posterior compartment and induced *tkv^CA*-expressing clones. However, no changes to actin enrichment or smoothness of clonal outlines could be observed in *bsk^DN*, *tkv^CA*-expressing clones (*Figure 4—figure supplement 2E-F*). Combined, these observations demonstrate that elevated actomyosin contractility acts upstream or in parallel to JNK interface signaling. Thus, JNK interface signaling may fulfill a different function in interface cells.

## JNK interface signaling drives cell elimination at clonal interfaces

Because JNK signaling can act as a promoter of apoptosis, we asked if JNK interface signaling serves to promote the elimination of cells, specifically at the interface. To test this idea, we first analyzed the

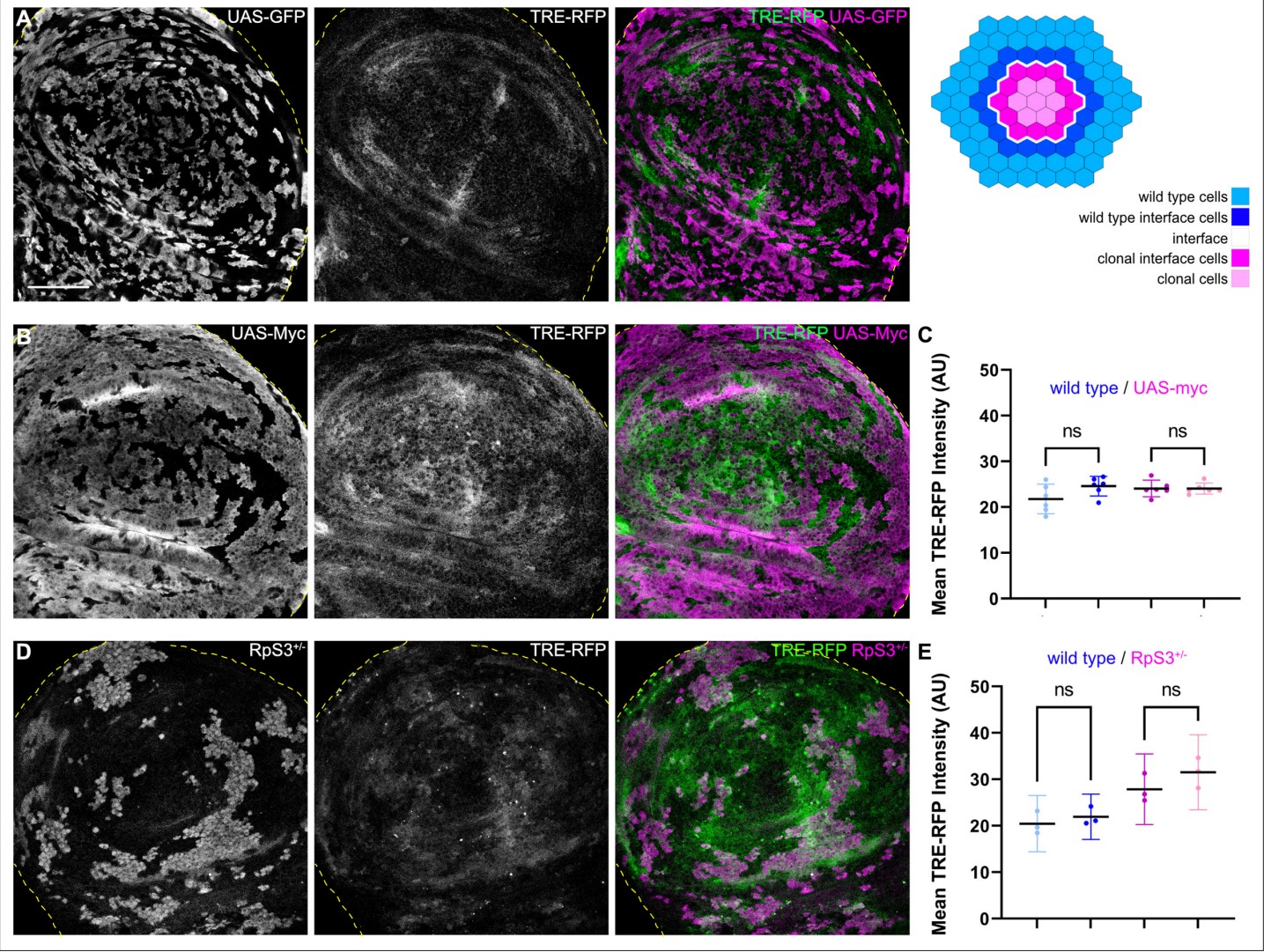

**Figure 4.** JNK interface signaling is unique to interface surveillance and is not activated by cell competition. (**A, B, and D**) Wing disc carrying mosaic clones (gray or magenta) expressing *GFP* (**A**), *Myc* (B, winner), or GFP-labeled clones heterozygous mutant for *RpS3* (D, loser). Discs express the JNK reporter TRE-RFP (gray or green). Yellow dashed lines demarcate the wing disc outline. (**C and E**) Quantifications of TRE-RFP intensities in specific zones of clones expressing *Myc* (**C**), or in clones heterozygous mutant for *RpS3* (**E**). See illustration above for color code of individual zones. Please note that in (**E**) wild type cells are only represented by an 4 μm band next to interface cells. This reduces the effect of highly variable and unpatterned activation of JNK in *RpS3* mosaic discs and distinguishes activation by a local interface mechanism from general JNK activity in wild-type cell row 2 distal to the interface. Graphs display mean ± 95% CI. One-way ANOVA tests were performed to test for statistical significance, ns = not significant. n=6 wing discs (**C**), n=3 wing discs (**E**). See *Figure 4—source data 1*; *Figure 4—source data 2*. Scale bars = 50 μm.

The online version of this article includes the following source data and figure supplement(s) for figure 4:

**Source data 1.** for *Figure 4C*.

**Source data 2.** for *Figure 4E*.

**Figure supplement 1.** JNK interface signaling is not activated by cell competition.

**Figure supplement 2.** JNK signaling is not required for actomyosin enrichment at clonal interfaces.

spatial distribution of apoptotic events in different mosaic genotypes (*fkh*, *ey*, and *tkv^{CA}*). To distinguish apoptotic events at the interface from apoptotic events in the clone interior, we specifically analyzed large clones with separable interface, interior, and exterior zones. This analysis revealed that, indeed, many apoptotic events occurred at clonal interfaces in *fkh*-, *ey*-, or *tkv^{CA}*-expressing clones. Importantly, apoptotic events were also elevated in wild type interface cells, indicating that

JNK interface signaling may also affect these wild-type cells if in contact with an aberrantly fated cell (*Figure 5A–E'*, *Figure 5—figure supplement 1A-C*).

To test if JNK signaling was indeed promoting apoptosis and elimination of aberrant cells specifically within the interface region, we asked how inhibition of JNK may change the spatial distribution of apoptotic events. We utilized two genetic strategies: we first inhibited JNK by expression of *bsk^DN* within *tkv^CA*-expressing clones. Indeed, intraclonal inhibition of JNK strongly reduced apoptosis at intraclonal interfaces (*Figure 5E–H*). Secondly, to also test the effect of JNK signaling in both clonal and wild-type interface cells, we expressed *bsk^DN* in the posterior compartment of a disc also harboring *tkv^CA* clones. We found that not only apoptosis at intraclonal interfaces in the posterior compartment was reduced but also apoptosis in wild-type interface cells, demonstrating that wild-type cell survival is also affected by JNK interface signaling (*Figure 5I–L*, *Figure 5—figure supplement 1D,D'*). The observation that both wild type and aberrant cell survival are regulated by JNK interface signaling supports our model that cell elimination by interface surveillance acts on differences in cellular fates rather than on a specific cell identity per se. Combined, these results clearly establish that JNK interface signaling strongly contributes to the elimination of cells at clonal interfaces and is thus a functional hallmark of the interface surveillance program.

Intriguingly, JNK interface signaling cannot account for all apoptotic events that we observed in aberrant clones. A second population of apoptotic cells, which could not be suppressed by expression of *bsk^DN*, located specifically to interior zones of clones (*Figure 5H and L*). These interior zones have lower JNK activity, but their apical surface often buckles to form cysts (*Bielmeier et al., 2016*). Theoretically, cysts arise when the contractile clonal interface compresses small and intermediate-sized clones, thereby driving apical surface buckling and cyst formation (*Bielmeier et al., 2016*). When we analyzed apoptotic patterns, we observed a strong spatial correlation between apical buckling and the apical presence of a subpopulation of apoptotic cells (*Figure 5—figure supplement 1E-G*). This raises the possibility that cell death in clone interiors may be driven by surface buckling dynamics and likely represents the source of JNK-independent apoptotic cell death. While this needs to be investigated further, we conclude that two spatially distinct and independent mechanisms act to promote elimination of aberrant cells.

## JNK interface signaling scales with clone size and specifically eliminates small clones

So how could a bilateral activation of pro-apoptotic JNK-signaling by both cell types create specificity for the elimination of aberrant cells? When a mutant cell arises, it is surrounded by wild-type cells. Thus, all surfaces of the aberrant cell - but only one surface in each wild type cell - will be in contact with a different fate (*Figure 6—figure supplement 1A*). As clone sizes increase and the topology of the interface changes, the number of contacts an aberrant cell has with wild-type cells will decrease, and clonal cells without interface contact will emerge. If the number of interface contacts may be viewed as the number of contacts that will trigger JNK activation, then JNK activity may generally scale with clone size. As predicted from this basic idea, we found that small *fkh*-expressing clones showed strongest overall activation of the JNK-reporter TRE-RFP (*Figure 6A*, *Figure 6—figure supplement 1B*). This finding mirrors the high levels of apoptosis observed in small *fkh*-expressing clones if compared to larger clones (see *Figure 1R*).

Importantly, these simplified topological considerations create separate predictions for how contacts of either interface cell will change with increasing clone size (*Figure 6—figure supplement 1A*): For enclosed (aberrant) interface cells, the number of contacts with surrounding (wild type) cells will be high in small clones and will strongly decrease with increasing clone size. In contrast: for surrounding (wild type) interface cells, the number of contacts with enclosed (aberrant) cells in small clones will be low, and this number will increase, although only mildly, with increasing clone size. To explore if the likelihood of JNK activation in either interface population may really scale with clone size, we specifically quantified JNK reporter intensities in each interface cell population. Indeed, JNK activation in *fkh*-expressing interface cells is initially higher and strongly decreases with increasing clone size (*Figure 6B*, *Figure 6—figure supplement 1C*). In contrast, JNK-activity in wild-type interface cells is overall low and only mildy increases with increasing clone size (*Figure 6C*, *Figure 6—figure supplement 1C*). This data suggests that cell contact topology may indeed be a determinant of bilateral JNK-activation strength.

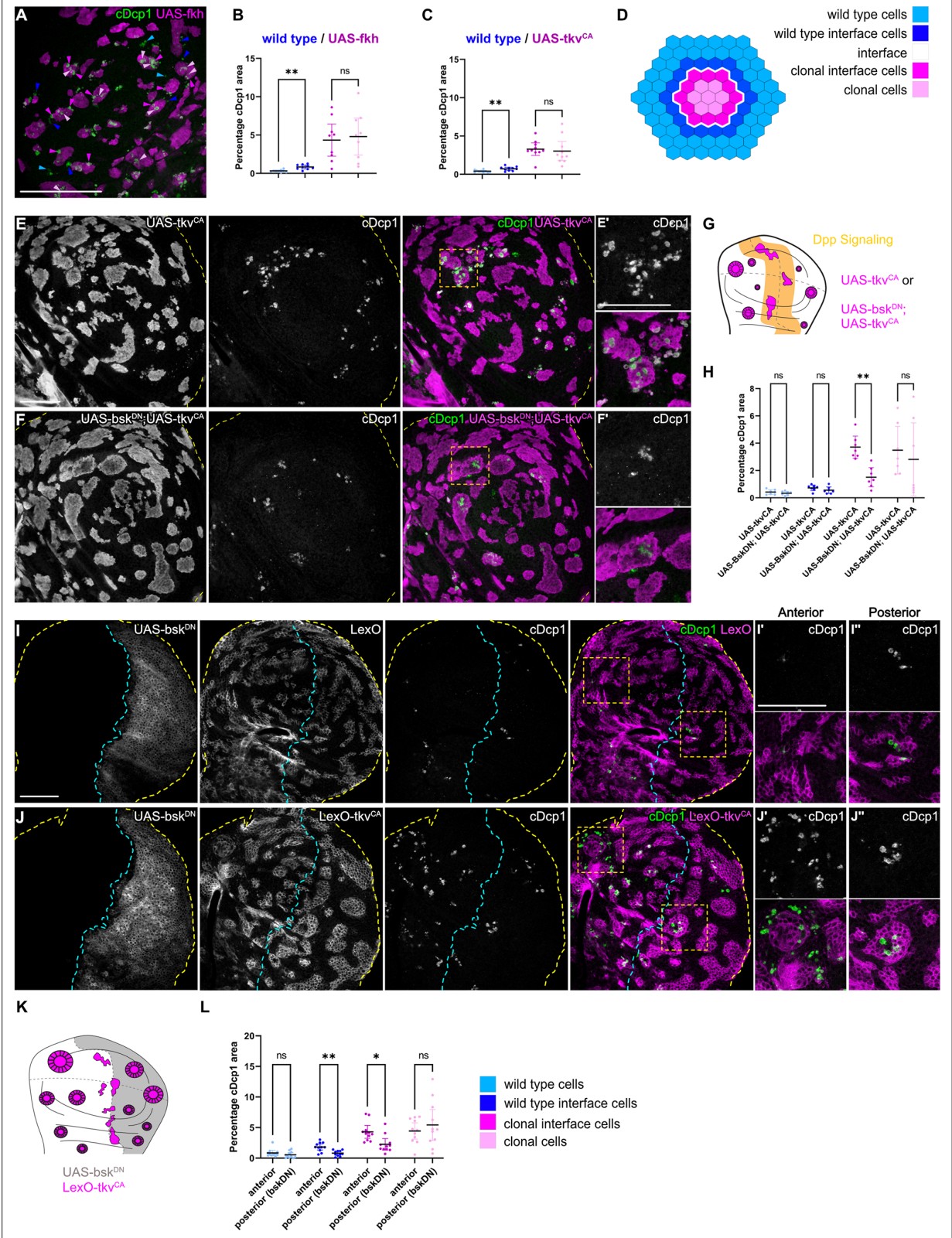

**Figure 5.** Cell elimination at clonal interfaces is mediated by JNK. (**A**) Maximum-intensity projection of basal sections from a wing imaginal disc carrying clones (magenta) expressing *fkh* and stained for cleaved Dcp1 (cDcp1) to visualize apoptosis (green). Colored arrows show apoptosis in different zones that were used for quantification: light blue (wild type cells), dark blue (wild type interface cells), magenta (clonal cells at interface), and light pink (clonal cells). Please see **Figure 5—figure supplement 1A** for zone segmentation. (**B–D**) Quantifications of relative percentage of apoptotic areas in selected

*Figure 5 continued on next page*

*Figure 5 continued*

zones around clones expressing UAS-*fkh* (**B**) or UAS-*tkv*$^{CA}$ (**C**). See (**D**) for color code of individual zones. Only large clones with a distinct interior zone of clonal cells were quantified. Graphs display mean ± 95% CI. One-way ANOVA tests were performed to test for statistical significance, ns = not significant, ** p≤0.01. n=9 wing discs (**B**) and n=10 wing discs (**C**). See *Figure 5—source data 1*. (**E–G**) Maximum-intensity projections of basal sections of wing imaginal discs carrying clones (gray or magenta) expressing *tkv*$^{CA}$ (**E**), or Bsk$^{DN}$,*tkv*$^{CA}$ (**F**) stained for cDcp1 to visualize apoptosis (gray or green). Yellow dashed lines demarcate the wing disc outline. Yellow frames mark regions shown in (**E' and F'**). Scheme (**G**) illustrates endogenous patterns of Dpp signaling (orange) and expected clone shape of *tkv*$^{CA}$ and *tkv*$^{CA}$, Bsk$^{DN}$ expressing clones (magenta) in areas where Dpp signaling is normally low (white). (**H**) Quantifications of relative percentage of apoptotic areas in selected zones around clones expressing expressing *tkv*$^{CA}$ or Bsk$^{DN}$,*tkv*$^{CA}$. See legend for color code. Only large clones with a distinct interior zone of clonal cells were quantified (see **A**). Graphs display mean ± 95% CI. Two-way ANOVA tests were performed to test for statistical significance, ns = not significant, * p≤0.05, ** p≤0.01. n=7 wing discs per genotype. See *Figure 5—source data 2*. (**I–K**): Maximum intensity projection of basal section of wing discs, where the posterior compartment expresses *Bsk*$^{DN}$ (gray) under the control of *en*-GAL4, and where clones (gray or magenta) express *LexO-mCherry* (**E**) or *LexO-tkv*$^{CA}$ (**F**) under the control of a LexA 'flip-out' system. Discs were stained for cDcp1 to visualize apoptosis (gray or green). Scheme in (**K**) illustrates the expected expression of *Bsk*$^{DN}$ in the posterior compartment and the response of *tkv*$^{CA}$-expressing clones (magenta). Yellow dashed lines demarcate the wing disc outline. Yellow frames mark regions selected in anterior (**I' and I'**) and posterior (**J'' and J''**) compartments. Dashed cyan line highlights the anterior-posterior compartment boundary. (**L**) Quantifications of relative percentage of apoptotic areas in selected zones around clones expressing LexO-*tkv*$^{CA}$ either in the anterior control compartment or in the posterior compartment expressing *Bsk*$^{DN}$. Only large clones with a distinct interior zone of clonal cells were quantified. Graphs display mean ± 95% CI. Two-way ANOVA tests were performed to test for statistical significance, ns = not significant, * p≤0.05, ** p≤0.01. n=11 wing discs. See *Figure 5—source data 3*. Scale bar = 50 μm.

The online version of this article includes the following source data and figure supplement(s) for figure 5:

**Source data 1.** for *Figure 5B*.

**Source data 2.** for *Figure 5H*.

**Source data 3.** for *Figure 5L*.

**Figure supplement 1.** Apoptosis can be observed at the buckling points of *fkh* expressing clones.

**Figure supplement 1—source data 1.** for *Figure 5—figure supplement 1B*.

**Figure supplement 1—source data 2.** for *Figure 5—figure supplement 1C*.

Lastly, we wanted to examine if the frequency of apoptosis in wild type or aberrant interface cells may scale with clone size and, consequently, with JNK-reporter activity. Indeed, apoptosis is high in *fkh*-expressing interface cells and lower in wild-type interface cells from small clones (*Figure 6D–G*). Moreover, we find that, as predicted, apoptosis in *fkh*-expressing interface cells decreases with increasing clone size (*Figure 6E*). Of note, we did not observe the mild increase of apoptosis in wild-type interface cells, as predicted for increasing clone sizes (*Figure 6F*). While apoptosis was still low, apoptosis was higher in wild-type interface cells around small clones than around large clones. We suggest that apoptosis of wild-type interface cell adjacent to small clones may be subject to additional apoptotic inputs, such as mechanical stress induced by buckling activity of clones, which is predicted to be higher in small clones than in larger clones (*Bielmeier et al., 2016*).

Combined, these results support a model where bilateral JNK activation is a read-out of clone-size-dependent cell contact topology, which may directly affect the frequency of apoptosis in interface contacting cells. The high number of interface contacts predicted for small aberrant clones would bias JNK activation toward aberrant interface cells and thus drive elimination of aberrant cells right when they appear in the tissue.

## Oncogenic *Ras*$^{V12}$ activates interface surveillance but suppresses apoptotic elimination

Mutations in Wnt, TGF-β, or cell fate specification pathways are widely reported to promote cancer (*Hiremath et al., 2022*; *Stuelten and Zhang, 2021*; *Brumbaugh et al., 2019*). We found that mutations in these pathways consistently elicit interface surveillance and, ultimately, apoptosis of affected cells. Thus, interface surveillance would play an important tumor suppressive role by facilitating elimination of cells with aberrant fate. One exception was revealed by our analysis of *Ras*$^{V12}$-expressing clones, representing an incredibly potent oncogenic mutations in cancer patients (*Moore et al., 2020*). *Ras*$^{V12}$ cells activate ERK signaling at high levels and exhibit ERK-pattern-specific actomyosin enrichment and JNK activation at the interface with surrounding wild-type cells (*Figure 7A–B''*; *Figure 7—figure supplement 1A-B'*, see *Figure 1—figure supplement 2H,N* for ERK-pattern). Thus, *Ras*$^{V12}$-expressing clones activate hallmarks of interface surveillance and are recognized as a distinct cell fate. However,

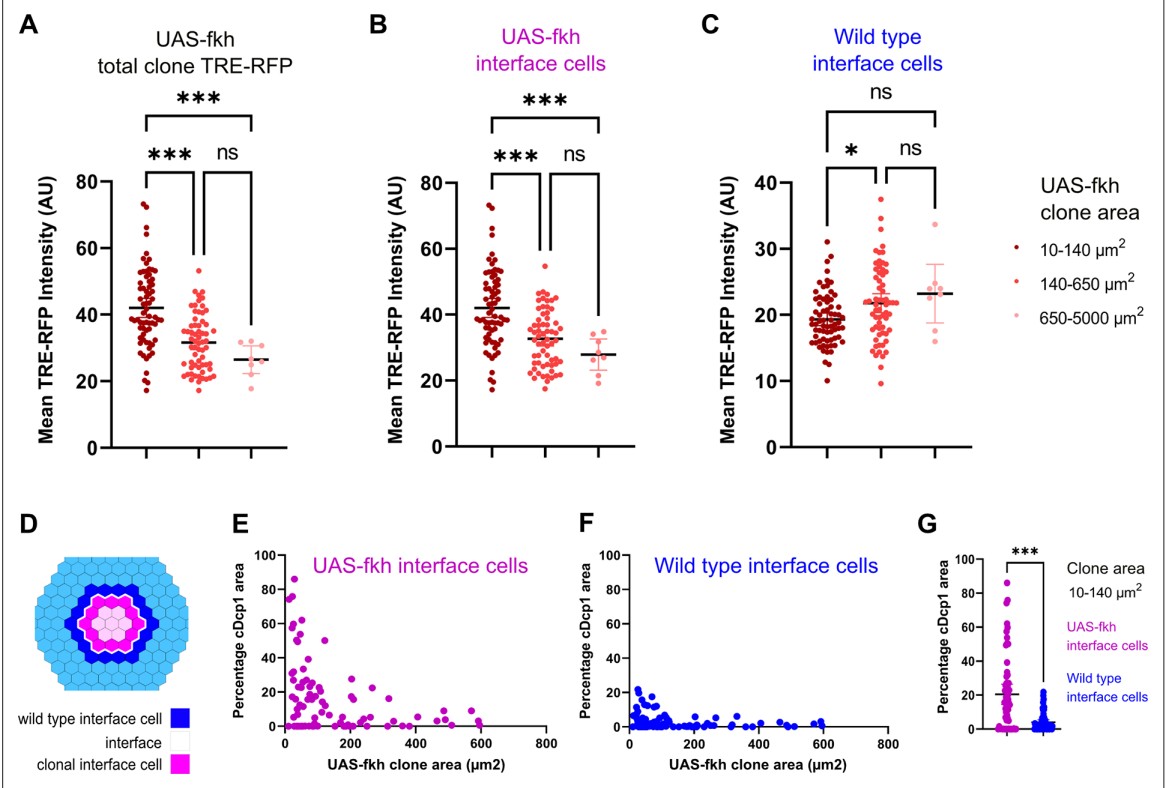

**Figure 6.** Bilateral JNK-signaling and apoptosis scale with clone size. (**A–C**) Quantification of TRE-RFP intensity within the entire clone area (**A**), only within clonal interface cells (**B**) and only within wild-type interface cells (**C**) in mosaic discs with *fkh*-expressing clones. Clones have been binned based on size: dark red (small clones, 10–140 µm², approximately containing 1–6 cells), red (medium clones, 140–650 µm², approximately containing 7–35 cells), and pink (large clones, 650–5000 µm², any clone above approximately 35 cells). A one-way ANOVA test was performed to test for statistical significance. n=140 clones from n=6 wing discs. Please note that JNK-activity within clonal interface cells decreases with clones size but increases with clones size within wild-type interface cells. This is consistent with expected changes in cell contact topology for clonal interface and wild-type interface cells when clone size increase (see *Figure 6—figure supplement 1A*, *Figure 6—source data 1*; *Figure 6—source data 2*; *Figure 6—source data 3*). (**D–G**) Scheme (**D**) depicting clonal interface and wild-type interface zones that were quantified in (**E–G**). Graph in (**E**) depicts distribution of the relative percentage of apoptotic area in clonal interface zones (**E**) and in wild-type interface zones (**F**) in different sizes of *fkh*-expressing clones. Note that wild-type interface cells are less apoptotic than clonal interface cells. This is consistent with lower levels of JNK-activity (compare B and C) arising from cell contact topologies for wild type and *fkh*-eypressing cells (see *Figure 6—figure supplement 1A*). Graph in (**G**) depicts the quantification of the relative percentage of apoptotic area in clonal interface zones (magenta) and in wild-type interface zones (blue) of small *fkh*-expressing clones (10–140 µm²) in the pouch of wing discs shown in (**E and F**). A one-way ANOVA test was performed to test for statistical significance. n=6 wing discs. See *Figure 6—source data 4*; *Figure 6—source data 5*; *Figure 6—source data 6*.

The online version of this article includes the following source data and figure supplement(s) for figure 6:

**Source data 1.** for *Figure 6A*.

**Source data 2.** for *Figure 6B*.

**Source data 3.** for *Figure 6C*.

**Source data 4.** for *Figure 6E*.

**Source data 5.** for *Figure 6F*.

**Source data 6.** for *Figure 6G*.

**Figure supplement 1.** Bilateral JNK-activity scales with clone size.

**Figure supplement 1—source data 1.** for *Figure 6—figure supplement 1B*.

**Figure supplement 1—source data 2.** for *Figure 6—figure supplement 1C*.

*Ras^V12*-expressing clones completely failed to induce apoptosis and were not eliminated from imaginal discs, suggesting that *Ras^V12* potently suppresses interface- and buckling-associated apoptosis (*Figure 7C and D*). Moreover, *Ras^V12* suppressed apoptosis in *fkh*- and *ey*-expressing clones, likely via dominantly elevating ERK activity (*Figure 7E–G*; *Figure 7—figure supplement 1*, *Figure 7—figure*

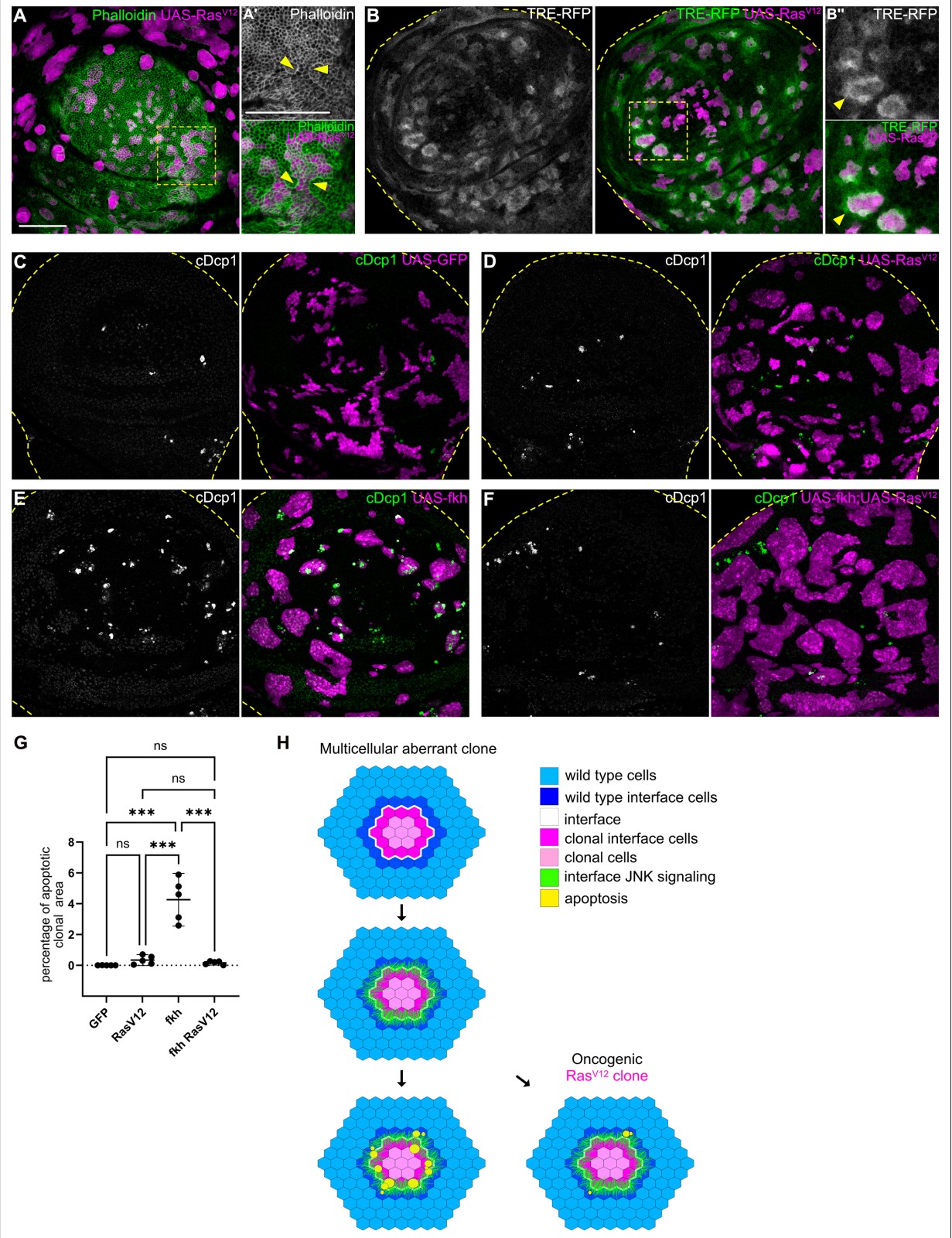

**Figure 7.** Cell elimination by interface surveillance is suppressed by oncogenic Ras$^{V12}$. (**A**) Wing disc carrying mosaic clones (magenta) expressing *Ras$^{V12}$* were stained with phalloidin to visualize Actin (gray or green). Yellow frames mark regions shown in (**A′**). Yellow arrows point to actin enrichment at clone boundaries. (**B**) Wing disc carrying mosaic clones (magenta) expressing *Ras$^{V12}$* and TRE-RFP (gray or green). Yellow dashed lines demarcate the wing disc outline. Yellow frames mark regions shown in (**B′**). Yellow arrows highlight TRE-RFP activation. Please compare JNK-activation pattern to pattern of

*Figure 7 continued on next page*

*Figure 7 continued*

endogenous EGF-activity in (*Figure 1—figure supplement 2H*). (**C, D, E, and F**) Maximum intensity projection of basal sections of wing discs carrying clones (magenta) expressing GFP (**C**) *Ras*$^{V12}$ (**D**), *fkh* (**E**), or *fkh*, *Ras*$^{V12}$ (**F**) was stained for cleaved Dcp1 (cDcp1) to visualize apoptosis (gray or green). Yellow dashed lines demarcate the wing disc outline. (**G**) Quantifications of the relative percentage of apoptotic area in clones expressing GFP only, *Ras*$^{V12}$; *fkh*, or *fkh*,*Ras*$^{V12}$. Graphs display mean ± 95% CI. One-way ANOVA tests were performed to test for statistical significance, ns = not significant, *** p≤0.001. n=5 wing discs per genotype. See *Figure 7—source data 1*. (**H**) Model of bilateral JNK activation and cell elimination by interface surveillance. Oncogenic mutations evade apoptosis. Scale bar = 50 μm.

The online version of this article includes the following source data and figure supplement(s) for figure 7:

**Source data 1.** for *Figure 7G*.

**Figure supplement 1.** Ras$^{V12}$ dominantly induces high-ERK signaling in *fkh*- and Ey-expressing clones.

**Figure supplement 1—source data 1.** for *Figure 7—figure supplement 1E* fkh.

**Figure supplement 1—source data 2.** for *Figure 7—figure supplement 1E–R* as-fkh.

**Figure supplement 1—source data 3.** for *Figure 7—figure supplement 1E–R* as.

**Figure supplement 2.** Ras$^{V12}$ rescues *ey*-expressing clones from apoptosis.

*supplement 2*). *Ras*$^{V12}$ is the only defined genotype for which we have observed complete evasion of apoptotic elimination by the interface surveillance program, highlighting the central role of oncogenic Ras as an initiating and cooperating factor in tumor formation. In conclusion, while misspecification of cell fate induces interface surveillance, oncogenic mutations may escape this tissue-intrinsic defense by promoting apoptosis resistance, thus increasing the likelihood of clone persistence and thus tumor growth.

## Discussion

Here, we demonstrate that cells expressing an aberrant fate or differentiation program induce activation of JNK at clonal interfaces (*Figure 7H*). Activation of JNK has been previously observed when steep differences in cell fates occur and has been suggested to be required for elimination of aberrant cells (*Monier and Suzanne, 2015*; *Manjón et al., 2007*; *Adachi-Yamada and O'Connor, 2002*; *Ohsawa et al., 2011*). We now define bilateral JNK signaling as a central new hallmark of interface surveillance and interface surveillance as a distinct branch of tissue-intrinsic error correction mechanisms protecting against cells with aberrant fate. Bilateral JNK activation is consistent with a model where interface surveillance monitors tissue health by assessing the degree of local differences. As difference per se does not carry information about which of the two contacting cells is aberrant, both cells respond by initiating stress-signaling via JNK activation. This is different than in cell-cell competition, where cell-autonomous activation of JNK signaling in loser cells due to deregulated cellular homeostasis predisposes the loser genotype to die (*Kucinski et al., 2017*). Indeed, bilateral JNK activation promotes apoptosis in cells of either genotype, also of wild type cells, that come into contact at the interface. Both JNK activation and apoptosis scale with the changing topology when clone sizes increase. In this interface topology model, bilateral JNK activation is particularly effective at eliminating single aberrant cells and small clones, which can indeed be observed experimentally (*Bielmeier et al., 2016*). This would reflect effective suppression of mutant cells by interface surveillance, as mutations in tissues affect single cells initially. While it may be surprising that evolution accepted that also wild-type cells are occasionally eliminated by bilateral JNK activity, the advantages of a such general detector of cell fate mispositioning may outweigh the disadvantage of collateral cell loss. Finally, we note that the topology of interface contacts between two large cell populations equalizes and reduces the number of potential JNK-activating interface contacts (*Figure 6—figure supplement 1A*). As a consequence, large clones are less likely to activate apoptosis and die. The clone size-dependent topology of bilateral JNK activation may thus also protect large fate compartments and their boundaries from high, potentially deleterious effects of JNK activation during development.

Strikingly, expression of oncogenic *Ras*$^{V12}$ induces hallmarks of interface surveillance, such as interface actomyosin, apical buckling, cyst formation (*Bielmeier et al., 2016*), as well as JNK interface signaling. However, *Ras*$^{V12}$ cells completely evade apoptosis, highlighting the potent oncogenic survival provided by *Ras*$^{V12}$ cooperativity. Importantly, expression of oncogenic Ras induces similar hallmark in mammalian MDCK cell culture or organoids, such as enrichment of actomyosin at cellular

interfaces (*Hogan et al., 2009*; *Kajita et al., 2014*; *Ohoka et al., 2015*) and elimination efficiencies that depend clone-size (*Sasaki et al., 2018*; *Kon et al., 2017*). Indeed, JNK interface-like signaling has been reported for oncogenic Src-expressing cells in organoid cultures (*Krotenberg Garcia et al., 2021*). Importantly, oncogenic Ras is a potent suppressor of apoptosis in flies and mammalian tissues (*Watanabe et al., 2018*). This likely prevents apoptotic elimination of these cells by interface surveillance from imaginal discs and, interestingly, by mechanical-compression-induced apoptosis from the pupal notum (*Levayer et al., 2016*; *Moreno et al., 2019*). However, why are *Ras^V12* cells then reported to be eliminated from mammalian tissue? Importantly, different studies conclude that *Ras^V12* cells are eliminated from epithelial layers by a cell-death-independent mechanism that involves strong activation of interfacial actomyosin activity, driving mechanical squeezing out of the tissue layer (*Kajita et al., 2014*; *Ohoka et al., 2015*; *Sasaki et al., 2018*; *Gudipaty and Rosenblatt, 2017*; *Wu et al., 2014*; *Yamamoto et al., 2016*; *Mitchell and Rosenblatt, 2021*; *Fadul et al., 2021*). As *Ras^V12* protects cells form apoptosis in both fly and mammalian contexts, we suggest that these initially contradictory observations actually reveal that interface surveillance and EDAC are evolutionarily conserved expression of the same ancient tissue-intrinsic defense system that specifically acts against the occurrence of aberrant cell fate and specification programs.

# Materials and methods
## Fly genetics
A list of strains, detailed genotypes, and experimental conditions are provided in *Supplementary file 1*; *Supplementary file 2*. Briefly, all crosses were kept on standard media. FLP/FRT and 'GAL4/UAS flip-out' and 'LexA/LexO flip-out' mosaic experiments utilized heat-shock-driven expression of a flipase. The respective crosses were allowed to lay eggs for 72 hr at 25°C followed by a heat-shock at 37°C for 60 min (FLP/FRT) or 8–25 min ('flip-out'). Larvae were dissected at wandering third instar stage or as indicated (30 hr and 54 hr after heat-shock).

## Immunohistochemistry and imaging
Imaginal discs were dissected and fixed in 4% formaldehyde/PBS for 15 min at room temperature (RT). The samples were washed in PBS +0.1 Triton X-100 (PBT) and then incubated in PBT + 5% normal goat serum (PBTN) for 10 min for blocking. Discs were incubated with primary antibodies overnight at 4°C: rabbit anti-Cleaved *Drosophila* cDcp-1 (1:200, Cell Signaling, #9578 S), rabbit anti-*fkh* (1:200, gift from Martin Juenger), mouse anti-eye (1:100, DSHB anti-eye), mouse anti-wingless (1:100, DSHB #4D4-s), goat anti-Distalless (Santa Cruz Biotechnology Distal-less df-20), rat anti-Ci (1:50, DSHB #2A1-s), rat anti-RFP (1:1000, Chromotek #5F8), rabbit anti-RFP (1:200, MBL #PM005), chicken anti-mCherry (1:1000, Abcam #ab205402), rat anti-Drosophila E-cadherin (1:50, DSHB DCAD2-s), and chicken anti-GFP (1:1000, Abcam #ab13970). The discs were then washed in PBT, followed by another blocking step with PBTN. The samples were counterstained with: DAPI (0.25 ng/l, Sigma), Phalloidin (Abcam Phalloidin 405 ab176752 1:1000, Sigma Aldrich Phalloidin 555 P1951 1:400, Invitrogen Phalloidin 647 1:100), and secondary antibodies from Invitrogen, 1:500 (goat anti-rabbit A11008, goat anti-chicken A11039, goat anti-chicken A21437, goat anti-rat A21434, goat anti-mouse A32728, goat anti-rabbit A21244, goat anti-chicken A21449, goat anti-mouse A21235, and donkey anti-goat A32849) and incubated for 3 hr at RT. Discs were again washed in PBT and PBS, then mounted using Molecular Probes Antifade Reagents (#S2828). To prevent squeezing of samples by coverslips for imaging the apical architecture without interference from the peripodium, two stripes of double-sided tape (Tesa, #05338) were placed on the slide. Samples were imaged using a Leica SP8 confocal microscope. The figures were assembled in Affinity Design.

## Image analysis and statistics
Where possible, control, and experimental samples were fixed, processed and mounted together to ensure comparable staining and imaging conditions. The signals of the following fluorescent reporters were further amplified by anti-GFP or anti-RFP antibody staining: *TRE-RFP* and *miniCiC-mCherry*. Images were processed, analyzed, and quantified using tools in Fiji (ImageJ2.3.0/; *Schindelin et al., 2012*; see details below). Great care was taken to apply consistent methods (i.e. number of projected sections or thresholding methods) within experiments. Statistical tests were performed in GraphPad

PRISM9. Details and the number of wing discs used for each test (n) can be found in the respective figure legends. Figure panels were assembled using Affinity Design.

## Image segmentation and quantification

### Epithelial integration of clones

Z-projections of maximum intensity of 2–3 slices of apical and basal sections were generated. An ROI was defined around the central region of the pouch to avoid confounding results due to the folded structure of the wing disc in the hinge. The number of clones in the ROI in the pouch of the apical and basal sections were counted. A paired t-test was used to statistically compare the number of clones detected apically and basally.

### Quantification of apoptosis within clones

Z-projections of maximum intensity of 2–3 slices of basal sections were generated. To define the total region of the wing disc, a Gaussian blur filter of sigma = 4 was applied to the DAPI channel. Intensity-based thresholding using 'triangle dark' threshold function was then used to generate a DAPI-based wing disc mask. To create a mask of GFP-labeled clones, the variation in GFP intensities were first pre-processed by applying a maximum and minimum filter of radius = 2. Then, intensity-based thresholding using the 'default dark' threshold function was performed to create a binary image. 'Despeckle' and 'fill holes' functions as well as a size inclusion range of 10-infinity µm$^2$ were applied to create the final clonal mask. The wing disc mask and the clonal GFP mask were used to define (by Boolean functions) different ROIs and then extract parameters, such as total clonal area, whole disc area, and background area. To define the apoptotic area, the cDcp1 channel was pre-processed by applying maximum and minimum filters, each of radius = 1. An intensity threshold ('intermodes dark') for cDcp1 channel was set in such a way that only high intensity cDcp1 particles were picked up. cDcp1 particles were quantified in the different ROIs by 'limit to threshold function', and the percentage apoptotic area in each ROI was calculated. A paired t-test test was performed to test for statistical significance. In addition, the total area and apoptotic area were measured for individual clones for an analysis of clone-size-dependent apoptosis. A correlation analysis using the Spearman coefficient and a non-linear regression analysis were performed on the data set. A one-way ANOVA was performed to test for statistical significance between different clone area bins.

### Quantification of TRE-RFP reporter activity in interface surveillance and cell-cell competition models

Z-projections of maximum intensity of 2–3 slices of basal sections were generated. To define the total region of the wing disc, a Gaussian blur filter of radius = 4 was applied to the DAPI channel. Intensity-based thresholding using 'minimum dark' threshold function was then used to generate a DAPI-based wing disc mask. To create a mask of GFP-labeled clones, the variation in GFP intensities were first pre-processed by applying a maximum and minimum filter of radius = 2. Then, intensity-based thresholding using the 'otsu' threshold function was performed to create a binary image. 'Despeckle' and 'fill holes' functions as well as a size inclusion range of 10-infinity µm$^2$ were applied to create the final clonal mask. To determine the average size of cells in the wing disc to define a 1 cell wide ROIs for wild type and clonal interfaces, we measured the size of 30 nuclei in three wild-type wing discs and determined their average size as 3.75 µm. To account for cytoplasm in the densely packed pseudostratified tissue, we set the size of each ROI to be 4 µm wide. The enlarge function was used on the GFP clonal mask to generate a 4 µm thick band around clones, representing the wild-type interface ROI (dark blue) as an approximately 1 cell thick band. Then, a second mask of only large clones was generated by defining an ROI with clones of a minimum area of 180 µm$^2$ to visualize and only select clones that also contained an 'interior clonal cells (light pink)' population. On this second mask of large clones, we applied the enlarge function of –4 µm to create 'clonal interface cells' ROI (magenta) and the 'interior clonal cells' ROI (light pink). Using Boolean functions on these ROIs, the wing disc and the clonal mask, all ROIs required for further analysis could be generated. For example, to generate the background wild-type cell ROI, clones and their wild type interface regions were excluded from the whole disc. Ultimately, the fluorescence intensity of the TRE-RFP channel was measured in different ROIs. Please note that for the quantification of the background TRE-RFP intensity in the discs with RpS3$^{-/-}$ clones, the background was defined as a 4 µm band of wild-type cells adjacent to the wild-type interface cells,

as we observed a general upregulation of JNK-signaling independent from an interface pattern in the mosaic RpS3$^{-/-}$ wing discs. One-way ANOVA tests were performed.

## Quantification of TRE-RFP reporter activity in the center and periphery of the pouch in clones expressing GFP or UAS-*tkv*$^{CA}$ clones

A single representative section of the disc proper was selected, which contained nuclei of both clonal cells and wild-type cells and used for further analysis. To create a mask of GFP-labeled clones, the variation in GFP intensities were first pre-processed by applying a maximum and minimum filter of radius = 2. Then, intensity-based thresholding using the 'otsu' threshold function was performed to create a binary image, and a size inclusion range of 10-infinity µm$^2$ was applied to create the final clonal mask. Individual clones were then picked in such a way that a 4 µm band could be drawn around them using the enlarge function, without this region encroaching into the area of surrounding clones. The criteria for picking different clones were as follows: *Dad*-LacZ intensity in clones in the center of the pouch and in the 4 micron band around the respective clone was in the range of ±20 AU; however, the difference in intensity in *Dad*-lacZ clones and the 4 micron band around the clone was much larger in the pouch periphery (the Dad-LacZ intensity in the 4 micron band around clones in pouch periphery was close to 0). The channel with TRE-RFP signal was selected, and the fluorescence intensity in the clones in the center and periphery of the pouch was measured. A paired t-test was used to statistically analyze the TRE-RFP reporter intensity in clones expressing GFP or *tkv*$^{CA}$ in the center and the periphery of the pouch.

## Quantification of percentage of apoptotic area in the presence and absence of JNK signaling

The percentage of apoptotic area in wing discs expressing aberrant clones was analyzed in in presence (5B, 5C S5B, and S5C) and in the absence of JNK signaling (5 H,L). We used two different model systems to study the effect of JNK inhibition on the apoptosis of aberrant clones. We first suppressed intra-clonal JNK activity only, for example, flip-out clones co-expressing UAS-*tkv*$^{CA}$ and UAS-bsk$^{DN}$ (5 H). For the second model, we suppressed JNK activity in the entire posterior compartment using the en-GAL4-UAS expression system, and the *tkv*$^{CA}$ expressing clones were under the control of the LexA-LexO system (5 L).

Z-projections of maximum intensity of 5–6 slices of basal sections were generated. To define the total region of the wing disc, a Gaussian blur filter of radius = 4 was applied to the DAPI channel. Intensity-based thresholding using 'minimum dark' threshold function was then used to generate a DAPI-based wing disc mask. To create a mask of GFP-labeled clones, the variation in GFP intensities were first pre-processed by applying a maximum and minimum filter of radius = 2. Then, intensity-based thresholding using the 'otsu' threshold function was performed to create a binary image. 'Despeckle' and 'fill holes' functions as well as a size inclusion range of 10-infinity µm$^2$ were applied to create the final clonal mask. The enlarge function was used on the GFP clonal mask to generate a 4 µm thick band around clones, representing the wild-type interface ROI (dark blue) as an approximately 1 cell thick band around every clone. Then, a second mask of only large clones was generated by defining an ROI with clones of a minimum area of 180 µm$^2$ to visualize and only select clones that also contained an 'interior clonal cells (light pink)' population. On this second mask of large clones, we applied the enlarge function of –4 µm to create 'clonal interface cells' ROI (magenta) and the 'interior clonal cells' ROI (light pink). Additionally in 5 H, ROIs of the posterior and anterior wing disc compartments were generated using the en-GAL4,UAS-GFP channel and the Boolean function XOR, respectively. Using Boolean functions on these ROIs, the wing disc and the clonal mask, all ROIs required for further analysis could be generated. For example, to generate the background wild-type cell ROI, clones and their wild-type interface regions were excluded from the whole disc. To define the apoptotic area, the cDcp1 channel was pre-processed by applying maximum and minimum filters, each of radius = 1. An intensity threshold ('intermodes dark') for cDcp1 channel was set in such a way that only high intensity cDcp1 particles were picked up. cDcp1 particles were quantified in the different ROIs by 'limit to threshold function', and the percentage apoptotic area in each ROI was calculated. For measuring apoptosis of the wild-type cells at the wild-type cell interface and the remaining wild-type population ('background'), a cDcp1 mask was generated from the pre-processed and thresholded cDcp1 channel. A GFP mask was generated, which contained all GFP-positive particles. The image calculator

function 'AND' was used to generate a GFP-positive cDcp1 mask. Then, the image calculator function 'Subtract' was used between the cDcp1 mask and the GFP-positive cDcp1 mask to generate the mask of GFP-negative cDcp1 particles. An ROI of these GFP negative particles was created, and the AND function was employed between the wild-type interface ROI and the background ROIs to measure the area of GFP-negative cDcp1 in the respective ROIs. A one-way ANOVA or two-way ANOVA test was performed to test for statistical significance.

## Quantification of TRE intensities and apoptosis within individual clones

A single representative section of the disc proper was selected, which contained nuclei of both clonal and wild-type cells to measure TRE-RFP intensity. Z-projections of maximum intensity of 2–3 slices of basal sections were generated to measure apoptosis. To define the total region of the wing disc, a Gaussian blur filter of sigma = 4 was applied to the DAPI channel. Intensity-based thresholding using 'triangle dark' threshold function was then used to generate a DAPI-based wing disc mask. To create a mask of GFP-labeled clones, the variation in GFP intensities were first pre-processed by applying a maximum and minimum filter of radius = 2. Then, intensity-based thresholding using the 'default dark' threshold function was performed to create a binary image. 'Despeckle' and 'fill holes' functions as well as a size inclusion range of 10-infinity $\mu m^2$ were applied to create the final clonal mask. The enlarge function was used on the GFP clonal mask to generate a 4 $\mu m$ thick band around clones, representing the wild-type interface ROI (dark blue) as an approximately 1 cell thick band around every clone. Then, a second mask of only large clones was generated by defining an ROI with clones of a minimum area of 180 $\mu m^2$ in to visualize and only select clones that also contained an 'interior clonal cells (light pink)' population. On this second mask of large clones, we applied the enlarge function of –4 $\mu m$ to create 'clonal interface cells' ROI (magenta) and the 'interior clonal cells' ROI (light pink). Individual clones of varying sizes in the pouch and hinge were selected. While TRE-RFP and cDcp1 areas were measured for all ROIs in the medium and large clones, on the total clone TRE-RFP and cDcp1 areas, and wild-type interface data was measured for the small clones. The wing disc mask and the clonal GFP mask were used to define (by Boolean functions) different ROIs and then extract parameters, such as total clonal area, clonal interface area, and clonal and wild-type interface areas. To define the apoptotic area, the cDcp1 channel was pre-processed by applying maximum and minimum filters, each of radius = 1. An intensity threshold ('intermodes dark') for cDcp1 channel was set in such a way that only high intensity cDcp1 particles were picked up. cDcp1 particles were quantified in the different ROIs by 'limit to threshold function', and the percentage apoptotic area in each ROI was calculated. For measuring apoptosis of the wild-type cells at the wild-type cell interface and the remaining wild-type population ('background'), a cDcp1 mask was generated from the pre-processed and thresholded cDcp1 channel. A GFP mask was generated, which contained all GFP-positive particles. The image calculator function 'AND' was used to generate a GFP-positive cDcp1 mask. Then, the image calculator function 'Subtract' was used between the cDcp1 mask and the GFP-positive cDcp1 mask to generate the mask of GFP-negative cDcp1 particles. An ROI of these GFP-negative particles was created, and the AND function was employed between the wild-type cell interface ROI and the background ROIs to measure the area of GFP-negative cDcp1 in the respective ROIs. A correlation anaylsis using the Spearman coefficient and a non-linear semi-log regression analysis were performed on the data set of individual clone. The clones were also separated into bins according to their size, and a one-way ANOVA was performed to test for statistical significance.

## Analysis of spatial correlation between buckling and apoptosis

Maximum intensity projections were generated, which specifically excluded the peripodium, when analysis of all apoptotic events in 3D was required. The GFP-marked clonal mask was generated using intensity-based thresholding, followed the 'fill holes' and 'despeckle' functions to remove noise. Medium-sized clones with an area of buckling as well as a planar area were selected and added to the ROI manager. A whole disc cDcp1 mask was generated using intensity-based thresholding on the cDcp1 channel. A clonal cDcp1 mask was generated using Boolean function AND and added to the ROI manager. Maximum projections of the apical junctional network (E-cad) were used to outline the buckling area within a clone with the oval selection tool, which was then added to the ROI manager. Buckling was defined as a characteristic reduction in cell surface areas observed in max projections of the curved surface during buckling. Buckling was verified by analysis of the reprojected Z-sliced

stack. An area of similar size and shape was then selected in non-buckling, planar area of the same clone. Care was taken that the planar region was not close to a clonal boundary to avoid confounding effects of JNK-dependent interface signaling on apoptotic behavior. The clonal cDcp1 within the buckling area or the planar area was combined by means of the 'AND' function. Lastly, cDcp1 area measurements in both buckling and planar areas were performed. 11 clones from at least 3 different wing discs were quantified.

### Quantification of *miniCic*-reporter activity

A single representative section of the disc proper was selected from the stack for further analysis. A nuclear binary mask for the whole disc was obtained by intensity-based thresholding of the DAPI channel and added to the ROI manager. To create a binary mask of GFP-labeled clones, we used intensity-based thresholding, followed by the 'despeckle' function and 'fill holes' function. To compare the clonal *miniCic* signal with the signal in surrounding wild-type cells, individual clones were selected as and added to the ROI manager. An outer band ROI (4 µm, corresponding to one cell row) was established around the selected clones using the 'enlarge' function. An ROI of clonal nuclei was obtained by using the Boolean function AND on the nuclear mask and the respective clone ROI. Similarly, the outer band of nuclei ROI was also generated for the respective clones. Then the fluorescence signal intensity in the *miniCic* channel was measured for individual clones in the clonal nuclei ROI and the outer band nuclei ROI. The number of clones analyzed in each experiment is noted in the figure legends.

## Acknowledgements

We thank the staff of the Life Imaging Center (LIC) in the Hilde Mangold House (HMH) of the Albert-Ludwigs-University of Freiburg for help with their confocal microscopy resources and the excellent support in image recording. We specifically thank the DFG for supporting our imaging work through project number 414136422. We thank David Bilder, Dirk Bohmann, Suzanne Eaton, Iswar Hariharan, Martin Juenger, Romain Levayer, Giorgios Pyrowolakis, and Helena Richardson for sharing reagents. We thank the Bloomington *Drosophila* Stock Center (BDSC), the Vienna *Drosophila* Stock Collection (VDRC), and the Developmental Studies Hybridoma Bank (DSHB) for providing fly stocks and antibodies. We also thank the SGBM and IMPRS-IE graduate school for supporting our students.

Funding for this work was provided by the Deutsche Forschungsgemeinschaft (DFG, German Research Foundation) under Germany´s Excellence Strategy (CIBSS – EXC-2189 – Project ID 390939984 and GSC-4, Spemann Graduate School of Biology and Medicine), the CRC 850 (Control of Cell Motility in Development and Cancer, A08), the Heisenberg Program (CL490/3-1), and the Boehringer Ingelheim Foundation Plus3 Program.

## Additional information

### Funding

| Funder | Grant reference number | Author |
|---|---|---|
| Deutsche Forschungsgemeinschaft | CRC850/A08 | Anne-Kathrin Classen |
| Deutsche Forschungsgemeinschaft | CL490/3-1 | Anne-Kathrin Classen |
| Deutsche Forschungsgemeinschaft | CIBSS-390939984 | Anne-Kathrin Classen |
| Deutsche Forschungsgemeinschaft | SGBM-GSC-4 | Deepti Prasad |
| Boehringer Ingelheim Foundation | BIS Plus3 | Anne-Kathrin Classen |

The funders had no role in study design, data collection and interpretation, or the decision to submit the work for publication.

## Author contributions
Deepti Prasad, Conceptualization, Data curation, Formal analysis, Validation, Investigation, Visualization, Methodology, Writing – original draft, Writing – review and editing; Katharina Illek, Formal analysis, Validation, Investigation, Visualization, Methodology; Friedericke Fischer, Investigation, Methodology, Writing – review and editing; Katrin Holstein, Investigation, Methodology; Anne-Kathrin Classen, Conceptualization, Supervision, Funding acquisition, Visualization, Writing – original draft, Project administration, Writing – review and editing

## Author ORCIDs
Katrin Holstein ⬤ https://orcid.org/0000-0003-1540-8378
Anne-Kathrin Classen ⬤ http://orcid.org/0000-0001-5157-0749

## Decision letter and Author response
Decision letter https://doi.org/10.7554/eLife.80809.sa1
Author response https://doi.org/10.7554/eLife.80809.sa2

## Additional files

### Supplementary files
• Supplementary file 1. Fly strains. Table listing fly strains used in this study.
• Supplementary file 2. Detailed genotypes. Table listing detailed genotypes per figure panels
• MDAR checklist

### Data availability
All data generated or analysed during this study are included in the manuscript and supporting file; Source Data files have been provided for Figures 1, 2, 4, 5, 6 and S1, S2.1, S2.2, S5 and S6.1 .

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
