## [Editor Report]

In this work, Prasad et al. show in *Drosophila* imaginal discs that a process of interface surveillance removes cells with inappropriate cell fate via activation of *JNK* and apoptosis. Importantly, *JNK* activity and apoptosis target most efficiently cells at the interface of small clusters of aberrant cells, but cells expressing oncogenic mutations are refractory to elimination, suggesting a mechanism for tumor formation. Therefore, this study is of interest to both the cell and developmental biology and the cancer biology fields.

---

## [Decision Letter]

**Decision letter after peer review:**

Thank you for submitting your article "Bilateral *JNK* activation is a hallmark of interface contractility and promotes elimination of aberrant cells" for consideration by *eLife*. Your article has been reviewed by 3 peer reviewers, and the evaluation has been overseen by a Reviewing Editor and Jonathan Cooper as the Senior Editor. The reviewers have opted to remain anonymous.

Essential revisions:

1 – Quantitative characterization of much of the data is missing. Only single representative images are shown for many of the experiments. The manuscript would strengthen massively when these images are supported with a quantitative measurement. For example (but not limited to), TRE-GFP in correctly vs mis-specified clones in Figure 2K-L, TRE-GFP intensity in Figure 3, clonal analysis in Figure 5.

2 – The authors claim that removal of aberrant cells is dependent on clone size and only occurs when aberrant cells are the minority compared to the surrounding tissue. Currently, there is no data in the manuscript that supports this claim. The only analysis of tissues containing a majority of miss-specified cells (Figures 2I-2J) shows a bilateral activation of *JNK*, similar to a minority of miss-specified cells. To support the claim that the phenotype is size dependent further analysis of clone size in relation to apoptosis and *JNK* activation is essential.

3 – Range of *JNK* activation. It is difficult to evaluate the exact range of *JNK* activation on both side of the clones relative to cell diameter. It is however an important point especially if the mechanism is somehow based on direct contact between cells (as claimed by the authors). Could the author try to quantify more precisely the average *JNK* signal at different distance from clone boundary? Ideally, this could be done after apical surface cell segmentation to see to which extent it is restricted to one raw of cells on each side and using some local projection tools to restrict as much as possible the number of plane projected (for instance using the local projector plugin from Fiji).

4 – *JNK* and cell-autonomous regulation. The authors validate that expression of TRE-GFP is dependent on *JNK* signaling, through over-expression of a dominant negative variant of the *JNK* kinase (BSKDN) in clones of miss-specified cells (ey or tkv). This experiment nicely shows that activation of *JNK* in surrounding WT cells is not altered. This furthermore illustrates that *JNK* signaling in the miss-specified cells is not needed for activation of *JNK* in their neighbors. However, this does not support the conclusion that *JNK* is activated in a cell autonomous fashion in either of these populations. The interaction of the two cell types can still cause signaling, but through inhibition of one of the kinases within the pathway, this just does not lead to downstream activation of TRE-GFP. In fact, one could argue that the expression of TRE-GFP is not cell-autonomous, because tkvCA clones that are not mis-specified (within dad4-LacZ regions) do not show induction of TRE-GFP (Figure 2L). The only way to untangle cell autonomous vs non-autonomous effects is through manipulation of upstream communication between the different cell populations. Such experiments, for example, manipulation of contractility, are likely beyond the scope of this study. Therefore, I would suggest rephrasing this paragraph.

5 – Apoptosis. The authors use p35 to prevent apoptosis but p35-protected cells are themselves known to activate *JNK*. It would be better to use DIAP1 expression, or to add Dronc-DN to p35. Preventing cell death with p35+dronc-DN could possibly restore the loss of mis-specified cells, because p35-protected cells are known to produce Dronc-dependent growth signals, which could actually be what is maintaining these cells.

6 – This study in context. While this study investigates a wide range of genetic backgrounds and uses comprehensive approaches, there are precedents in the literature that should be mentioned and discussed. Specifically, previous studies showed similar patterns of *JNK* activity upon local distortion of morphogens (so called morphogenetic apoptosis, Adachi-Yamada and O'Connor Dev Biol 2002), or the pattern of *JNK* activation observed near polariy mutant clones (Ohsawa et al., Dev Cell 2011) suggesting that this bilateral *JNK* activation might not be completely unique to these contexts. Previously, the expression "morphogenetic apoptosis" was used by Adachi-Yamada, please acknowledge and discuss those previous studies and tone down assertions of novelty accordingly.

7 – Clarifications and accuracy of statements. There are several areas in which editorial revisions should be considered. The authors used the terminology "interfacial contractility" to define the elimination context that they previously characterized (most likely to delineate it from other competition scenarios and as this was one of first distinctive feature that they found). This may give the impression (which is probably not what the authors wants to convey) that the *JNK* activation pattern is a byproduct of the interfacial contractility, which the authors have not shown. Possibly, an unknown factor associated with all these mislocalisation contexts triggers both an increase of lateral tension and in parallel this pattern of *JNK*. Unless the authors can demonstrate that interfacial tension and sorting are causal to *JNK* activation, it would be best to re-phrase throughout the text, the title and the abstract (for instance, figure 2 title "Interfacial contractility activates *JNK* interface signaling") and use some other descriptor such as pattern defect.

8 – Clarifications and accuracy of statements. The model proposed by the authors at the end (relationship between number of contact and *JNK* levels) seems a bit premature and not fully supported by current data. If this was true, one would expect differences in the *JNK* levels in small isolated cells compared to boundary or larger clones, and it does not really seem to be the case (e.g.: Figure 2I, there are few UAS-Fkj cells in the center of the WT large patch on the left and they don't have higher levels of *JNK* compared to the UAS-fkh cells surrounding the WT patch and most likely sharing less contact). Actually the relationship between clone size and probability to disappear could be explained by several alternative models: (i) if cell death is mostly restricted to the boundary, while proliferation distributed throughout the clone, then the ratio death over cell division will scale with the perimeter over the surface ratio of the clone. As a result, clone above a certain size will not disappear. (ii) Alternatively, the size effect can be explained by the previous study of the authors. Assuming that part of the death is trigger by pressure coming from lateral contractility, then pressure should be higher for small clones (small curvature) and may lead to this preferential elimination. Therefore, unless the author can show a correlation between *JNK* levels and number of neighbors, it would be best to keep open the models explaining this size effect and certainly not include this in the main figure.

---

## [Author Response]

Essential revisions:1 – Quantitative characterization of much of the data is missing. Only single representative images are shown for many of the experiments. The manuscript would strengthen massively when these images are supported with a quantitative measurement. For example (but not limited to), TRE-GFP in correctly vs mis-specified clones in Figure 2K-L, TRE-GFP intensity in Figure 3, clonal analysis in Figure 5.

We agree and performed additional quantification experiments and modified some quantification workflows, as requested by others below. These new quantification data sets can now be found in:

– Figure 1 Q,R

– Figure 2 I-L

– Figure 3 G,H

– Figure 5 B,C

– Figure 6 A-G

– Figure 1 —figure supplement 3 A, E

– Figure 2 —figure supplement 1 B,D

– Figure 5 —figure supplement 1 B,C

– Figure 6 —figure supplement 1 B,C

We hope that these data sets convincingly strengthen the core messages of the paper.

2 – The authors claim that removal of aberrant cells is dependent on clone size and only occurs when aberrant cells are the minority compared to the surrounding tissue. Currently, there is no data in the manuscript that supports this claim. The only analysis of tissues containing a majority of miss-specified cells (Figures 2I-2J) shows a bilateral activation of JNK, similar to a minority of miss-specified cells. To support the claim that the phenotype is size dependent further analysis of clone size in relation to apoptosis and JNK activation is essential.

We performed this analysis which is now shown in new Figure 6 and new Figure 6 —figure supplement 1. This new data supports of our model that changes in cell contact topology due to changes in clone size alter *JNK* activation strength. In addition, however, we minimized the minority/majority analogy in our text, as this concept was useful to highlight the differences to cell competition but appeared too abstract and distracted from what we actually think is import, which is the interface shape.

3 – Range of JNK activation. It is difficult to evaluate the exact range of JNK activation on both side of the clones relative to cell diameter. It is however an important point especially if the mechanism is somehow based on direct contact between cells (as claimed by the authors). Could the author try to quantify more precisely the average JNK signal at different distance from clone boundary? Ideally, this could be done after apical surface cell segmentation to see to which extent it is restricted to one raw of cells on each side and using some local projection tools to restrict as much as possible the number of plane projected (for instance using the local projector plugin from Fiji).

We agree that segmentation of apical surfaces would be ideal for this analysis. We are aware of the local projector plugin for FIJI and have used it in Figure 5 —figure supplement 1E. However, the relatively small size of the apical cell surface and distortion of its shape due to mechanics arising from interface contraction, apical surface buckling or apoptosis makes segmentation and interpretation of the junctional network for quantitative purposes exceedingly difficult. In addition, the segmented apical network will not always correspond well with the more lateral cytoplasmic/nuclear signals for the *JNK* reporter. Our approach to quantify *JNK*-activity in a lateral section offers a reproducible spatial representation of cells and *JNK* reporter activity. To address your comment experimentally, we now provide an additional analysis in Figure 2 C, F, IK and Figure 2 —figure supplement 1 A-D: we introduced an extra band (ie. ‘cell row 2’) next to interface cells to better illustrate the significant drop in *JNK*-signaling next to the interface cell band. We hope that this modification improves the spatial description of the bilateral JNKsignaling.

4 – JNK and cell-autonomous regulation. The authors validate that expression of TRE-GFP is dependent on JNK signaling, through over-expression of a dominant negative variant of the JNK kinase (BSKDN) in clones of miss-specified cells (ey or tkv). This experiment nicely shows that activation of JNK in surrounding WT cells is not altered. This furthermore illustrates that JNK signaling in the miss-specified cells is not needed for activation of JNK in their neighbors. However, this does not support the conclusion that JNK is activated in a cell autonomous fashion in either of these populations. The interaction of the two cell types can still cause signaling, but through inhibition of one of the kinases within the pathway, this just does not lead to downstream activation of TRE-GFP. In fact, one could argue that the expression of TRE-GFP is not cell-autonomous, because tkvCA clones that are not mis-specified (within dad4-LacZ regions) do not show induction of TRE-GFP (Figure 2L). The only way to untangle cell autonomous vs non-autonomous effects is through manipulation of upstream communication between the different cell populations. Such experiments, for example, manipulation of contractility, are likely beyond the scope of this study. Therefore, I would suggest rephrasing this paragraph.

We completely agree and rephrased that entire section.

5 – Apoptosis. The authors use p35 to prevent apoptosis but p35-protected cells are themselves known to activate JNK. It would be better to use DIAP1 expression, or to add Dronc-DN to p35. Preventing cell death with p35+dronc-DN could possibly restore the loss of mis-specified cells, because p35-protected cells are known to produce Dronc-dependent growth signals, which could actually be what is maintaining these cells.

We have now included new experiments where we use Diap1 to prevent cell death in aberrant clones. These experiments did not change our conclusion. The quantified data is now included in (Figure 1 —figure supplement 1). Please note that we performed and quantified similar experiments previously, which are published in Figure 6 A-I, Figure S6 J-N in reference [1].

6 – This study in context. While this study investigates a wide range of genetic backgrounds and uses comprehensive approaches, there are precedents in the literature that should be mentioned and discussed. Specifically, previous studies showed similar patterns of JNK activity upon local distortion of morphogens (so called morphogenetic apoptosis, Adachi-Yamada and O'Connor Dev Biol 2002), or the pattern of JNK activation observed near polariy mutant clones (Ohsawa et al., Dev Cell 2011) suggesting that this bilateral JNK activation might not be completely unique to these contexts. Previously, the expression "morphogenetic apoptosis" was used by Adachi-Yamada, please acknowledge and discuss those previous studies and tone down assertions of novelty accordingly.

We apologize for this lack of citation. Please allow us to explain:

We are now commenting on and including Adachi-Yamada and O'Connor and Ohsawa et al. in the beginning of the discussion, together with Monier and Suzanne and Manjon et al.

Of note, we have not been able to reproduce bilateral *JNK* activation in *scrib* mutant clones (Ohsawa et al) possibly because of differences in allelic strength and are thus refraining from discussing *scrib* mutant clones.

Also, we did not want to use the term ‘morphogenetic apoptosis’ for two reason: interface apoptosis and interface contractility are parallel pathways of the same process and are induced not just by morphogen deregulation but also other fate specification processes. Secondly, the term morphogenetic apoptosis (in our perception) is now more closely associated with the morphogenetic consequences of apoptotic extrusion for tissue fold formation in development, as described in leg discs (see *Monier and Suzanne* and *Manjon et al.)*. We did not want to create confusion in the literature (by using the term ‘morphogentic apoptosis’) about what we think the causes of interface apoptosis are (any aberrant fate, not just morphogen-induced) and what we think the consequences of interface apoptosis are (we do not observe morphogenetic changes in the wing discs that we can attribute to interface/morphogenetic apoptosis). All characteristic morphological changes created by aberrant clones can still be observed in our experiments when we inhibit apoptosis by Diap1, p35 or BskDN. Even the apical buckling and cyst formation from the clone interior does not depend on any morphogenetic forces from apoptosis, as buckling and cyst are equally well observed when apoptosis is inhibited (see [1-3] and this manuscript).

7 – Clarifications and accuracy of statements. There are several areas in which editorial revisions should be considered. The authors used the terminology "interfacial contractility" to define the elimination context that they previously characterized (most likely to delineate it from other competition scenarios and as this was one of first distinctive feature that they found). This may give the impression (which is probably not what the authors wants to convey) that the JNK activation pattern is a byproduct of the interfacial contractility, which the authors have not shown. Possibly, an unknown factor associated with all these mislocalisation contexts triggers both an increase of lateral tension and in parallel this pattern of JNK. Unless the authors can demonstrate that interfacial tension and sorting are causal to JNK activation, it would be best to re-phrase throughout the text, the title and the abstract (for instance, figure 2 title "Interfacial contractility activates JNK interface signaling") and use some other descriptor such as pattern defect.

We completely agree. We have now termed the process ‘interface surveillance’ which improves readability and clarity of this manuscript, and provides an open/neutral term for future discussion of its role in tissue-intrinsic error correction processes but also developmental processes (for example compartment boundary formation).

8 – Clarifications and accuracy of statements. The model proposed by the authors at the end (relationship between number of contact and JNK levels) seems a bit premature and not fully supported by current data. If this was true, one would expect differences in the JNK levels in small isolated cells compared to boundary or larger clones, and it does not really seem to be the case (e.g.: Figure 2I, there are few UAS-Fkj cells in the center of the WT large patch on the left and they don't have higher levels of JNK compared to the UAS-fkh cells surrounding the WT patch and most likely sharing less contact). Actually the relationship between clone size and probability to disappear could be explained by several alternative models: (i) if cell death is mostly restricted to the boundary, while proliferation distributed throughout the clone, then the ratio death over cell division will scale with the perimeter over the surface ratio of the clone. As a result, clone above a certain size will not disappear. (ii) Alternatively, the size effect can be explained by the previous study of the authors. Assuming that part of the death is trigger by pressure coming from lateral contractility, then pressure should be higher for small clones (small curvature) and may lead to this preferential elimination. Therefore, unless the author can show a correlation between JNK levels and number of neighbors, it would be best to keep open the models explaining this size effect and certainly not include this in the main figure.

We removed the model from the main figure and moved it into supplements to explain the basics of the topology concept (Figure 6 —figure supplement 1A). However, we now provide evidence in Figure 6 and Figure 6 —figure supplement 1 that *JNK* levels in wild type and aberrant interface cells scale with clone size in the expected (opposite) manner. Moreover, our experimentally observed distribution of apoptosis confirms this model as well. While we did not quantify the exact number of neighboring cells at the interface due to spatial limitations for segmentation (small cell size, cell network distortions, and apical surface distortions – particularly in interface cells around small clones). Yet, we are certain that our new data in Figure 6 provides a basis to discuss a simple possible concept of how interface surveillance induces a scaling of a bilateral response between two cell types and drives elimination of single aberrant cells.

References

1. Bielmeier, C., et al., Interface Contractility between Differently Fated Cells Drives Cell Elimination and Cyst Formation. Current Biology, 2016. 26(5): p. 563-574.

2. Shen, J. and C. Dahmann, Extrusion of cells with inappropriate Dpp signaling from *Drosophila* wing disc epithelia. Science, 2005. 307(5716): p. 1789-90.

3. Widmann, T.J. and C. Dahmann, Wingless signaling and the control of cell shape in

*Drosophila* wing imaginal discs. Dev Biol, 2009. 334(1): p. 161-73.

4. Stennicke, H.R., C.A. Ryan, and G.S. Salvesen, Reprieval from execution: the molecular basis of caspase inhibition. Trends Biochem Sci, 2002. 27(2): p. 94-101.